# Data-Dependent Coresets for Compressing Neural Networks with Applications to Generalization Bounds

**Cenk Baykal**[†*]**, Lucas Liebenwein**[†*]**, Igor Gilitschenski**[†]**, Dan Feldman**[‡]**, Daniela Rus**[†]

## Abstract

We present an efficient coresets-based neural network compression algorithm that sparsifies the parameters of a trained fully-connected neural network in a manner that provably approximates the network's output. Our approach is based on an importance sampling scheme that judiciously defines a sampling distribution over the neural network parameters, and as a result, retains parameters of high importance while discarding redundant ones. We leverage a novel, empirical notion of sensitivity and extend traditional coreset constructions to the application of compressing parameters. Our theoretical analysis establishes guarantees on the size and accuracy of the resulting compressed network and gives rise to generalization bounds that may provide new insights into the generalization properties of neural networks. We demonstrate the practical effectiveness of our algorithm on a variety of neural network configurations and real-world data sets.

## 1 Introduction

Within the past decade, large-scale neural networks have demonstrated unprecedented empirical success in high-impact applications such as object classification, speech recognition, computer vision, and natural language processing. However, with the ever-increasing size of state-of-the-art neural networks, the resulting storage requirements and performance of these models are becoming increasingly prohibitive in terms of both time and space. Recently proposed architectures for neural networks, such as those in Krizhevsky et al. (2012); Long et al. (2015); Badrinarayanan et al. (2015), contain millions of parameters, rendering them prohibitive to deploy on platforms that are resource-constrained, e.g., embedded devices, mobile phones, or small scale robotic platforms.

In this work, we consider the problem of sparsifying the parameters of a trained fully-connected neural network in a principled way so that the output of the compressed neural network is approximately preserved. We introduce a neural network compression approach based on identifying and removing weighted edges with low relative importance via coresets, small weighted subsets of the original set that approximate the pertinent cost function. Our compression algorithm hinges on extensions of the traditional sensitivity-based coresets framework (Langberg & Schulman, 2010; Braverman et al., 2016), and to the best of our knowledge, is the first to apply coresets to parameter downsizing. In this regard, our work aims to simultaneously introduce a practical algorithm for compressing neural network parameters with provable guarantees and close the research gap in prior coresets work, which has predominantly focused on compressing input data points.

In particular, this paper contributes the following:

1. A coreset approach to compressing problem-specific parameters based on a novel, empirical notion of sensitivity that extends state-of-the-art coreset constructions.

2. An efficient neural network compression algorithm, CoreNet, based on our extended coreset approach that sparsifies the parameters via importance sampling of weighted edges.

3. Extensions of the CoreNet method, CoreNet+ and CoreNet++, that improve upon the edge sampling approach by additionally performing neuron pruning and amplification.

[†]Computer Science and Artificial Intelligence Laboratory, Massachusetts Institute of Technology, emails: {`baykal, lucasl, igilitschenski, rus`}`@mit.edu`

[‡]Robotics and Big Data Laboratory, University of Haifa, email: `dannyf.post@gmail.com`

[*]These authors contributed equally to this work

4. Analytical results establishing guarantees on the approximation accuracy, size, and generalization of the compressed neural network.

5. Evaluations on real-world data sets that demonstrate the practical effectiveness of our algorithm in compressing neural network parameters and validate our theoretical results.

## 2 RELATED WORK

Our work builds upon the following prior work in coresets and compression approaches.

**Coresets** Coreset constructions were originally introduced in the context of computational geometry (Agarwal et al., 2005) and subsequently generalized for applications to other problems via an importance sampling-based, *sensitivity* framework (Langberg & Schulman, 2010; Braverman et al., 2016). Coresets have been used successfully to accelerate various machine learning algorithms such as $k$-means clustering (Feldman & Langberg, 2011; Braverman et al., 2016), graphical model training (Molina et al., 2018), and logistic regression (Huggins et al., 2016) (see the surveys of Bachem et al. (2017) and Munteanu & Schwiegelshohn (2018) for a complete list). In contrast to prior work, we generate coresets for reducing the number of parameters – rather than data points – via a novel construction scheme based on an efficiently-computable notion of sensitivity.

**Low-rank Approximations and Weight-sharing** Denil et al. (2013) were among the first to empirically demonstrate the existence of significant parameter redundancy in deep neural networks. A predominant class of compression approaches consists of using low-rank matrix decompositions, such as Singular Value Decomposition (SVD) (Denton et al., 2014), to approximate the weight matrices with their low-rank counterparts. Similar works entail the use of low-rank tensor decomposition approaches applicable both during and after training (Jaderberg et al., 2014; Kim et al., 2015; Tai et al., 2015; Ioannou et al., 2015; Alvarez & Salzmann, 2017; Yu et al., 2017). Another class of approaches uses feature hashing and weight sharing (Weinberger et al., 2009; Shi et al., 2009; Chen et al., 2015b;a; Ullrich et al., 2017). Building upon the idea of weight-sharing, quantization (Gong et al., 2014; Wu et al., 2016; Zhou et al., 2017) or regular structure of weight matrices was used to reduce the effective number of parameters (Zhao et al., 2017; Sindhwani et al., 2015; Cheng et al., 2015; Choromanska et al., 2016; Wen et al., 2016). Despite their practical effectiveness in compressing neural networks, these works generally lack performance guarantees on the quality of their approximations and/or the size of the resulting compressed network.

**Weight Pruning** Similar to our proposed method, weight pruning (LeCun et al., 1990) hinges on the idea that only a few dominant weights within a layer are required to approximately preserve the output. Approaches of this flavor have been investigated by Lebedev & Lempitsky (2016); Dong et al. (2017), e.g., by embedding sparsity as a constraint (Iandola et al., 2016; Aghasi et al., 2017; Lin et al., 2017). Another related approach is that of Han et al. (2015), which considers a combination of weight pruning and weight sharing methods. Nevertheless, prior work in weight pruning lacks rigorous theoretical analysis of the effect that the discarded weights can have on the compressed network. To the best of our knowledge, our work is the first to introduce a practical, sampling-based weight pruning algorithm with provable guarantees.

**Generalization** The generalization properties of neural networks have been extensively investigated in various contexts (Dziugaite & Roy, 2017; Neyshabur et al., 2017a; Bartlett et al., 2017). However, as was pointed out by Neyshabur et al. (2017b), current approaches to obtaining non-vacuous generalization bounds do not fully or accurately capture the empirical success of state-of-the-art neural network architectures. Recently, Arora et al. (2018) and Zhou et al. (2018) highlighted the close connection between compressibility and generalization of neural networks. Arora et al. (2018) presented a compression method based on the Johnson-Lindenstrauss (JL) Lemma (Johnson & Lindenstrauss, 1984) and proved generalization bounds based on succinct reparameterizations of the original neural network. Building upon the work of Arora et al. (2018), we extend our theoretical compression results to establish novel generalization bounds for fully-connected neural networks. Unlike the method of Arora et al. (2018), which exhibits guarantees of the compressed network's performance only on the set of training points, our method's guarantees hold (probabilistically) for any random point drawn from the distribution. In addition, we establish that our method can $\varepsilon$-approximate the neural network output neuron-wise, which is stronger than the norm-based guarantee of Arora et al. (2018).

In contrast to prior work, this paper addresses the problem of compressing a fully-connected neural network while *provably* preserving the network's output. Unlike previous theoretically-grounded compression approaches – which provide guarantees in terms of the normed difference –, our method provides the stronger entry-wise approximation guarantee, even for points outside of the available data

set. As our empirical results show, ensuring that the output of the compressed network entry-wise approximates that of the original network is critical to retaining high classification accuracy. Overall, our compression approach remedies the shortcomings of prior approaches in that it (i) exhibits favorable theoretical properties, (ii) is computationally efficient, e.g., does not require retraining of the neural network, (iii) is easy to implement, and (iv) can be used in conjunction with other compression approaches – such as quantization or Huffman coding – to obtain further improved compression rates.

# 3  PROBLEM DEFINITION

## 3.1  FULLY-CONNECTED NEURAL NETWORKS

A feedforward fully-connected neural network with $L \in \mathbb{N}_+$ layers and parameters $\theta$ defines a mapping $f_\theta : \mathcal{X} \to \mathcal{Y}$ for a given input $x \in \mathcal{X} \subseteq \mathbb{R}^d$ to an output $y \in \mathcal{Y} \subseteq \mathbb{R}^k$ as follows. Let $\eta^\ell \in \mathbb{N}_+$ denote the number of neurons in layer $\ell \in [L]$, where $[L] = \{1, \ldots, L\}$ denotes the index set, and where $\eta^1 = d$ and $\eta^L = k$. Further, let $\eta = \sum_{\ell=2}^L \eta^\ell$ and $\eta^* = \max_{\ell \in \{2, \ldots, L\}} \eta^\ell$. For layers $\ell \in \{2, \ldots, L\}$, let $W^\ell \in \mathbb{R}^{\eta^\ell \times \eta^{\ell-1}}$ be the weight matrix for layer $\ell$ with entries denoted by $w_{ij}^\ell$, rows denoted by $w_i^\ell \in \mathbb{R}^{1 \times \eta^{\ell-1}}$, and $\theta = (W^2, \ldots, W^L)$. For notational simplicity, we assume that the bias is embedded in the weight matrix. Then for an input vector $x \in \mathbb{R}^d$, let $a^1 = x$ and $z^\ell = W^\ell a^{\ell-1} \in \mathbb{R}^{\eta^\ell}$, $\forall \ell \in \{2, \ldots, L\}$, where $a^{\ell-1} = \phi(z^{\ell-1}) \in \mathbb{R}^{\eta^{\ell-1}}$ denotes the activation. We consider the activation function to be the Rectified Linear Unit (ReLU) function, i.e., $\phi(\cdot) = \max\{\cdot, 0\}$ (entry-wise, if the input is a vector). The output of the network for an input $x$ is $f_\theta(x) = z^L$, and in particular, for classification tasks the prediction is $\operatorname{argmax}_{i \in [k]} f_\theta(x)_i = \operatorname{argmax}_{i \in [k]} z_i^L$.

## 3.2  NEURAL NETWORK CORESET PROBLEM

Consider the setting where a neural network $f_\theta(\cdot)$ has been trained on a training set of independent and identically distributed (i.i.d.) samples from a joint distribution on $\mathcal{X} \times \mathcal{Y}$, yielding parameters $\theta = (W^2, \ldots, W^L)$. We further denote the input points of a validation data set as $\mathcal{P} = \{x_i\}_{i=1}^n \subseteq \mathcal{X}$ and the marginal distribution over the input space $\mathcal{X}$ as $\mathcal{D}$. We define the size of the parameter tuple $\theta$, $\mathrm{nnz}(\theta)$, to be the sum of the number of non-zero entries in the weight matrices $W^2, \ldots, W^L$.

For any given $\varepsilon, \delta \in (0, 1)$, our overarching goal is to generate a reparameterization $\hat{\theta}$, yielding the neural network $f_{\hat{\theta}}(\cdot)$, using a randomized algorithm, such that $\mathrm{nnz}(\hat{\theta}) \ll \mathrm{nnz}(\theta)$, and the neural network output $f_\theta(x)$, $x \sim \mathcal{D}$ can be approximated up to $1 \pm \varepsilon$ multiplicative error with probability greater than $1 - \delta$. We define the $1 \pm \varepsilon$ multiplicative error between two $k$-dimensional vectors $a, b \in \mathbb{R}^k$ as the following entry-wise bound: $a \in (1 \pm \varepsilon)b \Leftrightarrow a_i \in (1 \pm \varepsilon)b_i \, \forall i \in [k]$, and formalize the definition of an $(\varepsilon, \delta)$-coreset as follows.

**Definition 1** (($\varepsilon, \delta$)-coreset). *Given user-specified $\varepsilon, \delta \in (0, 1)$, a set of parameters $\hat{\theta} = (\hat{W}^2, \ldots, \hat{W}^L)$ is an ($\varepsilon, \delta$)-coreset for the network parameterized by $\theta$ if for $x \sim \mathcal{D}$, it holds that*

$$\mathbb{P}_{\hat{\theta}, x} \left( f_{\hat{\theta}}(x) \in (1 \pm \varepsilon) f_\theta(x) \right) \geq 1 - \delta,$$

*where $\mathbb{P}_{\hat{\theta}, x}$ denotes a probability measure with respect to a random data point $x$ and the output $\hat{\theta}$ generated by a randomized compression scheme.*

# 4  METHOD

In this section, we introduce our neural network compression algorithm as depicted in Alg. 1. Our method is based on an important sampling-scheme that extends traditional sensitivity-based coreset constructions to the application of compressing parameters.

## 4.1  CORENET

Our method (Alg. 1) hinges on the insight that a validation set of data points $\mathcal{P} \overset{i.i.d.}{\sim} \mathcal{D}^n$ can be used to approximate the relative importance, i.e., sensitivity, of each weighted edge with respect to the input

data distribution $\mathcal{D}$. For this purpose, we first pick a subsample of the data points $\mathcal{S} \subseteq \mathcal{P}$ of appropriate size (see Sec. 5 for details) and cache each neuron's activation and compute a neuron-specific constant to be used to determine the required edge sampling complexity (Lines 2-6).

---

**Algorithm 1** CORENET

---

**Input:** $\varepsilon, \delta \in (0,1)$: error and failure probability, respectively; $\mathcal{P} \subseteq \mathcal{X}$: a set of $n$ points from the input space $\mathcal{X}$ such that $\mathcal{P} \overset{i.i.d.}{\sim} \mathcal{D}^n$; $\theta = (W^2, \ldots, W^L)$: parameters of the original uncompressed neural network.
**Output:** $\hat{\theta} = (\hat{W}^2, \ldots, \hat{W}^L)$: sparsified parameter set such that $f_{\hat{\theta}}(\cdot) \in (1 \pm \varepsilon) f_{\theta}(\cdot)$ (see Sec. 5 for details).

1: $\varepsilon' \leftarrow \frac{\varepsilon}{2(L-1)}$; $\quad \eta^* \leftarrow \max_{\ell \in \{2,\ldots,L-1\}} \eta^{\ell}$; $\quad \eta \leftarrow \sum_{\ell=2}^{L} \eta^{\ell}$; $\quad \lambda^* \leftarrow \log(\eta\,\eta^*)/2$;
2: $\mathcal{S} \leftarrow$ Uniform sample (without replacement) of $\lceil \log(8\,\eta\,\eta^*/\delta) \log(\eta\,\eta^*) \rceil$ points from $\mathcal{P}$;
3: $a^1(x) \leftarrow x \quad \forall x \in \mathcal{S}$;
4: **for** $x \in \mathcal{S}$ **do**
5: $\quad$ **for** $\ell \in \{2, \ldots, L\}$ **do**
6: $\quad\quad a^{\ell}(x) \leftarrow \phi(W^{\ell} a^{\ell-1}(x)); \quad \Delta_i^{\ell}(x) \leftarrow \frac{\sum_{k \in [\eta^{\ell-1}]} |w_{ik}^{\ell} a_k^{\ell-1}(x)|}{\left| \sum_{k \in [\eta^{\ell-1}]} w_{ik}^{\ell} a_k^{\ell-1}(x) \right|}$;

7: **for** $\ell \in \{2, \ldots, L\}$ **do**
8: $\quad \hat{\Delta}^{\ell} \leftarrow \left( \frac{1}{|\mathcal{S}|} \max_{i \in [\eta^{\ell}]} \sum_{x \in \mathcal{S}} \Delta_i^{\ell}(x) \right) + \kappa$, where $\kappa = \sqrt{2\lambda_*} \left( 1 + \sqrt{2\lambda_*} \log(8\,\eta\,\eta^*/\delta) \right)$;
9: $\quad \hat{W}^{\ell} \leftarrow (\vec{0}, \ldots, \vec{0}) \in \mathbb{R}^{\eta^{\ell} \times \eta^{\ell-1}}$; $\quad \hat{\Delta}^{\ell \rightarrow} \leftarrow \prod_{k=\ell}^{L} \hat{\Delta}^k$; $\quad \varepsilon_{\ell} \leftarrow \frac{\varepsilon'}{\hat{\Delta}^{\ell \rightarrow}}$;
10: $\quad$ **for all** $i \in [\eta^{\ell}]$ **do**
11: $\quad\quad \mathcal{W}_+ \leftarrow \{j \in [\eta^{\ell-1}] : w_{ij}^{\ell} > 0\}; \quad \mathcal{W}_- \leftarrow \{j \in [\eta^{\ell-1}] : w_{ij}^{\ell} < 0\}$;
12: $\quad\quad \hat{w}_i^{\ell+} \leftarrow \text{SPARSIFY}(\mathcal{W}_+, w_i^{\ell}, \varepsilon_{\ell}, \delta, \mathcal{S}, a^{\ell-1}); \quad \hat{w}_i^{\ell-} \leftarrow \text{SPARSIFY}(\mathcal{W}_-, -w_i^{\ell}, \varepsilon_{\ell}, \delta, \mathcal{S}, a^{\ell-1})$;
13: $\quad\quad \hat{w}_i^{\ell} \leftarrow \hat{w}_i^{\ell+} - \hat{w}_i^{\ell-}; \quad \hat{W}_{i\bullet}^{\ell} \leftarrow \hat{w}_i^{\ell}; \quad\quad \triangleright$ Consolidate the weights into the $i^{\text{th}}$ row of $\hat{W}^{\ell}$;
14: **return** $\hat{\theta} = (\hat{W}^2, \ldots, \hat{W}^L)$;

---

**Algorithm 2** SPARSIFY$(\mathcal{W}, w, \varepsilon, \delta, \mathcal{S}, a(\cdot))$

---

**Input:** $\mathcal{W} \subseteq [\eta^{\ell-1}]$: index set; $w \in \mathbb{R}^{1 \times \eta^{\ell-1}}$: row vector corresponding to the weights incoming to node $i \in [\eta^{\ell}]$ in layer $\ell \in \{2, \ldots, L\}$; $\varepsilon, \delta \in (0,1)$: error and failure probability, respectively; $\mathcal{S} \subseteq \mathcal{P}$: subsample of the original point set; $a(\cdot)$: cached activations of previous layer for all $x \in \mathcal{S}$.
**Output:** $\hat{w}$: sparse weight vector.

1: **for** $j \in \mathcal{W}$ **do**
2: $\quad s_j \leftarrow \max_{x \in \mathcal{S}} \frac{w_j a_j(x)}{\sum_{k \in \mathcal{W}} w_k a_k(x)}$; $\quad\quad\quad\quad \triangleright$ Compute the sensitivity of each edge

3: $S \leftarrow \sum_{j \in \mathcal{W}} s_j$;
4: **for** $j \in \mathcal{W}$ **do** $\quad\quad\quad\quad \triangleright$ Generate the importance sampling distribution over the incoming edges
5: $\quad q_j \leftarrow \frac{s_j}{S}$;
6: $m \leftarrow \left\lceil \frac{8 S \log(\eta\,\eta^*) \log(8\,\eta/\delta)}{\varepsilon^2} \right\rceil$; $\quad\quad\quad\quad \triangleright$ Compute the number of required samples
7: $\mathcal{C} \leftarrow$ a multiset of $m$ samples from $\mathcal{W}$ where each $j \in \mathcal{W}$ is sampled with probability $q_j$;
8: $\hat{w} \leftarrow (0, \ldots, 0) \in \mathbb{R}^{1 \times \eta^{\ell-1}}$; $\quad\quad\quad\quad \triangleright$ Initialize the compressed weight vector
9: **for** $j \in \mathcal{C}$ **do** $\quad\quad \triangleright$ Update the entries of the sparsified weight matrix according to the samples $\mathcal{C}$
10: $\quad \hat{w}_j \leftarrow \hat{w}_j + \frac{w_j}{m\,q_j}$; $\quad\quad \triangleright$ Entries are reweighted by $\frac{1}{m\,q_j}$ to ensure unbiasedness of our estimator

11: **return** $\hat{w}$;

---

Subsequently, we apply our core sampling scheme to sparsify the set of incoming weighted edges to each neuron in all layers (Lines 7-13). For technical reasons (see Sec. 5), we perform the sparsification on the positive and negative weighted edges separately and then consolidate the results (Lines 11-13). By repeating this procedure for all neurons in every layer, we obtain a set $\hat{\theta} = (\hat{W}^2, \ldots, \hat{W}^L)$ of sparse weight matrices such that the output of each layer and the entire network is approximately preserved, i.e., $\hat{W}^{\ell} \hat{a}^{\ell-1}(x) \approx W^{\ell} a^{\ell-1}(x)$ and $f_{\hat{\theta}}(x) \approx f_{\theta}(x)$, respectively[1].

---

[1]$\hat{a}^{\ell-1}(x)$ denotes the approximation from previous layers for an input $x \sim \mathcal{D}$; see Sec. 5 for details.

## 4.2 SPARSIFYING WEIGHTS

The crux of our compression scheme lies in Alg. 2 (invoked twice on Line 12, Alg. 1) and in particular, in the importance sampling scheme used to select a small subset of edges of high importance. The cached activations are used to compute the *sensitivity*, i.e., relative importance, of each considered incoming edge $j \in \mathcal{W}$ to neuron $i \in [\eta^\ell]$, $\ell \in \{2, \ldots, L\}$ (Alg. 2, Lines 1-2). The relative importance of each edge $j$ is computed as the maximum (over $x \in \mathcal{S}$) ratio of the edge's contribution to the sum of contributions of all edges. In other words, the sensitivity $s_j$ of an edge $j$ captures the highest (relative) impact $j$ had on the output of neuron $i \in [\eta^\ell]$ in layer $\ell$ across all $x \in \mathcal{S}$.

The sensitivities are then used to compute an importance sampling distribution over the incoming weighted edges (Lines 4-5). The intuition behind the importance sampling distribution is that if $s_j$ is high, then edge $j$ is more likely to have a high impact on the output of neuron $i$, therefore we should keep edge $j$ with a higher probability. $m$ edges are then sampled with replacement (Lines 6-7) and the sampled weights are then reweighed to ensure unbiasedness of our estimator (Lines 9-10).

## 4.3 EXTENSIONS: NEURON PRUNING AND AMPLIFICATION

In this subsection we outline two improvements to our algorithm that that do not violate any of our theoretical properties and may improve compression rates in practical settings.

**Neuron pruning (CoreNet+)** Similar to removing redundant edges, we can use the empirical activations to gauge the importance of each neuron. In particular, if the maximum activation (over all evaluations $x \in \mathcal{S}$) of a neuron is equal to 0, then the neuron – along with all of the incoming and outgoing edges – can be pruned without significantly affecting the output with reasonable probability. This intuition can be made rigorous under the assumptions outlined in Sec. 5.

**Amplification (CoreNet++)** Coresets that provide stronger approximation guarantees can be constructed via *amplification* – the procedure of constructing multiple approximations (coresets) $(\hat{w}_i^\ell)_1, \ldots, (\hat{w}_i^\ell)_\tau$ over $\tau$ trials, and picking the best one. To evaluate the quality of each approximation, a different subset $\mathcal{T} \subseteq \mathcal{P} \setminus \mathcal{S}$ can be used to infer performance. In practice, amplification would entail constructing multiple approximations by executing Line 12 of Alg. 1 and picking the one that achieves the lowest relative error on $\mathcal{T}$.

## 5 ANALYSIS

In this section, we establish the theoretical guarantees of our neural network compression algorithm (Alg. 1). The full proofs of all the claims presented in this section can be found in the Appendix.

### 5.1 PRELIMINARIES

Let $x \sim \mathcal{D}$ be a randomly drawn input point. We explicitly refer to the pre-activation and activation values at layer $\ell \in \{2, \ldots, \ell\}$ with respect to the input $x \in \text{supp}(\mathcal{D})$ as $z^\ell(x)$ and $a^\ell(x)$, respectively. The values of $z^\ell(x)$ and $a^\ell(x)$ at each layer $\ell$ will depend on whether or not we compressed the previous layers $\ell' \in \{2, \ldots, \ell\}$. To formalize this interdependency, we let $\hat{z}^\ell(x)$ and $\hat{a}^\ell(x)$ denote the respective quantities of layer $\ell$ when we replace the weight matrices $W^2, \ldots, W^\ell$ in layers $2, \ldots, \ell$ by $\hat{W}^2, \ldots, \hat{W}^\ell$, respectively.

For the remainder of this section (Sec. 5) we let $\ell \in \{2, \ldots, L\}$ be an arbitrary layer and let $i \in [\eta^\ell]$ be an arbitrary neuron in layer $\ell$. For purposes of clarity and readability, we will omit the the variable denoting the layer $\ell \in \{2, \ldots, L\}$, the neuron $i \in [\eta^\ell]$, and the incoming edge index $j \in [\eta^{\ell-1}]$, whenever they are clear from the context. For example, when referring to the intermediate value of a neuron $i \in [\eta^\ell]$ in layer $\ell \in \{2, \ldots, L\}$, $z_i^\ell(x) = \langle w_i^\ell, \hat{a}^{\ell-1}(x) \rangle \in \mathbb{R}$ with respect to a point $x$, we will simply write $z(x) = \langle w, a(x) \rangle \in \mathbb{R}$, where $w := w_i^\ell \in \mathbb{R}^{1 \times \eta^{\ell-1}}$ and $a(x) := a^{\ell-1}(x) \in \mathbb{R}^{\eta^{\ell-1} \times 1}$. Under this notation, the weight of an incoming edge $j$ is denoted by $w_j \in \mathbb{R}$.

### 5.2 IMPORTANCE SAMPLING BOUNDS FOR POSITIVE WEIGHTS

In this subsection, we establish approximation guarantees under the assumption that the weights are positive. Moreover, we will also assume that the input, i.e., the activation from the previous layer, is

non-negative (entry-wise). The subsequent subsection will then relax these assumptions to conclude that a neuron's value can be approximated well even when the weights and activations are not all positive and non-negative, respectively. Let $\mathcal{W} = \{j \in [\eta^{\ell-1}] : w_j > 0\} \subseteq [\eta^{\ell-1}]$ be the set of indices of incoming edges with strictly positive weights. To sample the incoming edges to a neuron, we quantify the relative importance of each edge as follows.

**Definition 2** (Relative Importance). *The importance of an incoming edge $j \in \mathcal{W}$ with respect to an input $x \in \mathrm{supp}(\mathcal{D})$ is given by the function $g_j(x)$, where $g_j(x) = \frac{w_j \, a_j(x)}{\sum_{k \in \mathcal{W}} w_k \, a_k(x)} \quad \forall j \in \mathcal{W}$.*

Note that $g_j(x)$ is a function of the random variable $x \sim \mathcal{D}$. We now present our first assumption that pertains to the Cumulative Distribution Function (CDF) of the relative importance random variable.

**Assumption 1.** *For all $j \in \mathcal{W}$, the CDF of the random variable $g_j(x)$, denoted by $F_j(\cdot)$, satisfies*

$$F_j(M/K) \le \exp(-1/K), \quad \text{where } M = \min\{x \in [0,1] : F_j(x) = 1\},$$

*and $K \in [2, \log(\eta \, \eta^*)]$ is a universal constant.*[2]

Assumption 1 is a technical assumption on the ratio of the weighted activations that will enable us to rule out pathological problem instances where the relative importance of each edge cannot be well-approximated using a small number of data points $\mathcal{S} \subseteq \mathcal{P}$. Henceforth, we consider a uniformly drawn (without replacement) subsample $\mathcal{S} \subseteq \mathcal{P}$ as in Line 2 of Alg. 1, where $|\mathcal{S}| = \lceil \log(8 \, \eta \, \eta^*/\delta) \log(\eta \, \eta^*) \rceil$, and define the sensitivity of an edge as follows.

**Definition 3** (Empirical Sensitivity). *Let $\mathcal{S} \subseteq \mathcal{P}$ be a subset of distinct points from $\mathcal{P} \stackrel{i.i.d.}{\sim} \mathcal{D}^n$. Then, the sensitivity over positive edges $j \in \mathcal{W}$ directed to a neuron is defined as $s_j = \max_{x \in \mathcal{S}} g_j(x)$.*

Our first lemma establishes a core result that relates the weighted sum with respect to the sparse row vector $\hat{w}$, $\sum_{k \in \mathcal{W}} \hat{w}_k \, \hat{a}_k(x)$, to the value of the of the weighted sum with respect to the ground-truth row vector $w$, $\sum_{k \in \mathcal{W}} w_k \, \hat{a}_k(x)$. We remark that there is randomness with respect to the randomly generated row vector $\hat{w}_i^\ell$, a randomly drawn input $x \sim \mathcal{D}$, and the function $\hat{a}(\cdot) = \hat{a}^{\ell-1}(\cdot)$ defined by the randomly generated matrices $\hat{W}^2, \ldots, \hat{W}^{\ell-1}$ in the previous layers. Unless otherwise stated, we will henceforth use the shorthand notation $\mathbb{P}(\cdot)$ to denote $\mathbb{P}_{\hat{w}^\ell, \, x, \, \hat{a}^{\ell-1}}(\cdot)$. Moreover, for ease of presentation, we will first condition on the event $\mathcal{E}_{1/2}$ that $\hat{a}(x) \in (1 \pm 1/2) a(x)$ holds. This conditioning will simplify the preliminary analysis and will be removed in our subsequent results.

**Lemma 1** (Positive-Weights Sparsification). *Let $\varepsilon, \delta \in (0,1)$, and $x \sim \mathcal{D}$. $\mathrm{SPARSIFY}(\mathcal{W}, w, \varepsilon, \delta, \mathcal{S}, a(\cdot))$ generates a row vector $\hat{w}$ such that*

$$\mathbb{P}\left( \sum_{k \in \mathcal{W}} \hat{w}_k \, \hat{a}_k(x) \notin (1 \pm \varepsilon) \sum_{k \in \mathcal{W}} w_k \, \hat{a}_k(x) \; \Big| \; \mathcal{E}_{1/2} \right) \le \frac{3\delta}{8\eta}$$

*where $\mathrm{nnz}(\hat{w}) \le \left\lceil \frac{8 \, S \, \log(\eta \, \eta^*) \log(8 \, \eta/\delta)}{\varepsilon^2} \right\rceil$, and $S = \sum_{j \in \mathcal{W}} s_j$.*

### 5.3 Importance Sampling Bounds

We now relax the requirement that the weights are strictly positive and instead consider the following index sets that partition the weighted edges: $\mathcal{W}_+ = \{j \in [\eta^{\ell-1}] : w_j > 0\}$ and $\mathcal{W}_- = \{j \in [\eta^{\ell-1}] : w_j < 0\}$. We still assume that the incoming activations from the previous layers are positive (this assumption can be relaxed as discussed in Appendix A.2.4). We define $\Delta_i^\ell(x)$ for a point $x \sim \mathcal{D}$ and neuron $i \in [\eta^\ell]$ as $\Delta_i^\ell(x) = \frac{\sum_{k \in [\eta^{\ell-1}]} |w_{ik}^\ell \, a_k^{\ell-1}(x)|}{\left| \sum_{k \in [\eta^{\ell-1}]} w_{ik}^\ell \, a_k^{\ell-1}(x) \right|}$. The following assumption serves a similar purpose as does Assumption 1 in that it enables us to approximate the random variable $\Delta_i^\ell(x)$ via an empirical estimate over a small-sized sample of data points $\mathcal{S} \subseteq \mathcal{P}$.

**Assumption 2** (Subexponentiality of $\Delta_i^\ell(x)$). *For any layer $\ell \in \{2, \ldots, L\}$ and neuron $i \in [\eta^\ell]$, the centered random variable $\Delta = \Delta_i^\ell(x) - \mathbb{E}_{x \sim \mathcal{D}}[\Delta_i^\ell(x)]$ is subexponential (Vershynin, 2016) with parameter $\lambda \le \log(\eta \, \eta^*)/2$, i.e., $\mathbb{E}[\exp(s\Delta)] \le \exp(s^2 \lambda^2) \quad \forall |s| \le \frac{1}{\lambda}$.*[2]

---

[2]The upper bound of $\log(\eta\eta^*)$ for $K$ and $\lambda$ can be considered somewhat arbitrary in the sense that, more generally, we only require that $K, \lambda \in \mathcal{O}(\mathrm{polylog}(\eta\eta^*|\mathcal{P}|)$. Defining the upper bound in this way simplifies the presentation of the core ideas without having to deal with the constants involved in the asymptotic notation.

For $\varepsilon \in (0,1)$ and $\ell \in \{2, \ldots, L\}$, we let $\varepsilon' = \frac{\varepsilon}{2(L-1)}$ and define $\varepsilon_\ell = \frac{\varepsilon'}{\hat{\Delta}^{\ell\to}} = \frac{\varepsilon}{2(L-1)\prod_{k=\ell}^{L}\hat{\Delta}^k}$, where $\hat{\Delta}^\ell = \left(\frac{1}{|\mathcal{S}|}\max_{i\in[\eta^\ell]}\sum_{x'\in\mathcal{S}}\Delta_i^\ell(x')\right) + \kappa$. To formalize the interlayer dependencies, for each $i \in [\eta^\ell]$ we let $\mathcal{E}_i^\ell$ denote the (desirable) event that $\hat{z}_i^\ell(x) \in (1 \pm 2(\ell-1)\varepsilon_{\ell+1})z_i^\ell(x)$ holds, and let $\mathcal{E}^\ell = \cap_{i\in[\eta^\ell]}\mathcal{E}_i^\ell$ be the intersection over the events corresponding to each neuron in layer $\ell$.

**Lemma 2** (Conditional Neuron Value Approximation). *Let $\varepsilon, \delta \in (0,1)$, $\ell \in \{2, \ldots, L\}$, $i \in [\eta^\ell]$, and $x \sim \mathcal{D}$. CORENET generates a row vector $\hat{w}_i^\ell = \hat{w}_i^{\ell+} - \hat{w}_i^{\ell-} \in \mathbb{R}^{1\times\eta^{\ell-1}}$ such that*

$$\mathbb{P}\left(\mathcal{E}_i^\ell \mid \mathcal{E}^{\ell-1}\right) = \mathbb{P}\left(\hat{z}_i^\ell(x) \in (1 \pm 2(\ell-1)\varepsilon_{\ell+1})z_i^\ell(x) \mid \mathcal{E}^{\ell-1}\right) \geq 1 - \delta/\eta, \tag{1}$$

*where $\varepsilon_\ell = \frac{\varepsilon'}{\hat{\Delta}^{\ell\to}}$ and $\mathrm{nnz}(\hat{w}_i^\ell) \leq \left\lceil \frac{8\,S\,\log(\eta\,\eta^*)\log(8\,\eta/\delta)}{\varepsilon_\ell^2}\right\rceil + 1$, where $S = \sum_{j\in\mathcal{W}_+}s_j + \sum_{j\in\mathcal{W}_-}s_j$.*

The following core result establishes unconditional layer-wise approximation guarantees and culminates in our main compression theorem.

**Lemma 3** (Layer-wise Approximation). *Let $\varepsilon, \delta \in (0,1)$, $\ell \in \{2, \ldots, L\}$, and $x \sim \mathcal{D}$. CORENET generates a sparse weight matrix $\hat{W}^\ell \in \mathbb{R}^{\eta^\ell\times\eta^{\ell-1}}$ such that, for $\hat{z}^\ell(x) = \hat{W}^\ell\hat{a}^\ell(x)$,*

$$\mathbb{P}_{(\hat{W}^2,\ldots,\hat{W}^\ell),\,x}\left(\mathcal{E}^\ell\right) = \mathbb{P}_{(\hat{W}^2,\ldots,\hat{W}^\ell),\,x}\left(\hat{z}^\ell(x) \in (1 \pm 2(\ell-1)\varepsilon_{\ell+1})z^\ell(x)\right) \geq 1 - \frac{\delta\sum_{\ell'=2}^{\ell}\eta^{\ell'}}{\eta}.$$

**Theorem 4** (Network Compression). *For $\varepsilon, \delta \in (0,1)$, Algorithm 1 generates a set of parameters $\hat{\theta} = (\hat{W}^2, \ldots, \hat{W}^L)$ of size*

$$\mathrm{nnz}(\hat{\theta}) \leq \sum_{\ell=2}^{L}\sum_{i=1}^{\eta^\ell}\left(\left\lceil \frac{32(L-1)^2(\hat{\Delta}^{\ell\to})^2 S_i^\ell \log(\eta\,\eta^*)\log(8\,\eta/\delta)}{\varepsilon^2}\right\rceil + 1\right)$$

*in $\mathcal{O}\left(\eta\,\eta^*\log\left(\eta\,\eta^*/\delta\right)\right)$ time such that $\mathbb{P}_{\hat{\theta},\,x\sim\mathcal{D}}\left(f_{\hat{\theta}}(x) \in (1\pm\varepsilon)f_\theta(x)\right) \geq 1 - \delta$.*

We note that we can obtain a guarantee for a set of $n$ randomly drawn points by invoking Theorem 4 with $\delta' = \delta/n$ and union-bounding over the failure probabilities, while only increasing the sampling complexity logarithmically, as formalized in Corollary 12, Appendix A.2.

## 5.4 Generalization Bounds

As a corollary to our main results, we obtain novel generalization bounds for neural networks in terms of empirical sensitivity. Following the terminology of Arora et al. (2018), the expected margin loss of a classifier $f_\theta : \mathbb{R}^d \to \mathbb{R}^k$ parameterized by $\theta$ with respect to a desired margin $\gamma > 0$ and distribution $\mathcal{D}$ is defined by $L_\gamma(f_\theta) = \mathbb{P}_{(x,y)\sim\mathcal{D}_{\mathcal{X},\mathcal{Y}}}\left(f_\theta(x)_y \leq \gamma + \max_{i\neq y}f_\theta(x)_i\right)$. We let $\hat{L}_\gamma$ denote the empirical estimate of the margin loss. The following corollary follows directly from the argument presented in Arora et al. (2018) and Theorem 4.

**Corollary 5** (Generalization Bounds). *For any $\delta \in (0,1)$ and margin $\gamma > 0$, Alg. 1 generates weights $\hat{\theta}$ such that with probability at least $1 - \delta$, the expected error $L_0(f_{\hat{\theta}})$ with respect to the points in $\mathcal{P} \subseteq \mathcal{X}$, $|\mathcal{P}| = n$, is bounded by*

$$L_0(f_{\hat{\theta}}) \leq \hat{L}_\gamma(f_\theta) + \widetilde{\mathcal{O}}\left(\sqrt{\frac{\max_{x\in\mathcal{P}}\|f_\theta(x)\|_2^2\,L^2\sum_{\ell=2}^{L}(\hat{\Delta}^{\ell\to})^2\sum_{i=1}^{\eta^\ell}S_i^\ell}{\gamma^2\,n}}\right).$$

## 6 Results

In this section, we evaluate the practical effectiveness of our compression algorithm on popular benchmark data sets (*MNIST* (LeCun et al., 1998), *FashionMNIST* (Xiao et al., 2017), and *CIFAR-10* (Krizhevsky & Hinton, 2009)) and varying fully-connected trained neural network configurations: 2 to 5 hidden layers, 100 to 1000 hidden units, either fixed hidden sizes or decreasing hidden size denoted by *pyramid* in the figures. We further compare the effectiveness of our sampling scheme in

reducing the number of non-zero parameters of a network, i.e., in sparsifying the weight matrices, to that of uniform sampling, Singular Value Decomposition (SVD), and current state-of-the-art sampling schemes for matrix sparsification (Drineas & Zouzias, 2011; Achlioptas et al., 2013; Kundu & Drineas, 2014), which are based on matrix norms – $\ell_1$ and $\ell_2$ (Frobenius). The details of the experimental setup and results of additional evaluations may be found in Appendix B.

**Experiment Setup**  We compare against three variations of our compression algorithm: (i) sole edge sampling (CoreNet), (ii) edge sampling with neuron pruning (CoreNet+), and (iii) edge sampling with neuron pruning and amplification (CoreNet++). For comparison, we evaluated the average relative error in output ($\ell_1$-norm) and average drop in classification accuracy relative to the accuracy of the uncompressed network. Both metrics were evaluated on a previously unseen test set.

**Results**  Results for varying architectures and datasets are depicted in Figures 1 and 2 for the average drop in classification accuracy and relative error ($\ell_1$-norm), respectively. As apparent from Figure 1, we are able to compress networks to about 15% of their original size without significant loss of accuracy for networks trained on *MNIST* and *FashionMNIST*, and to about 50% of their original size for *CIFAR*.

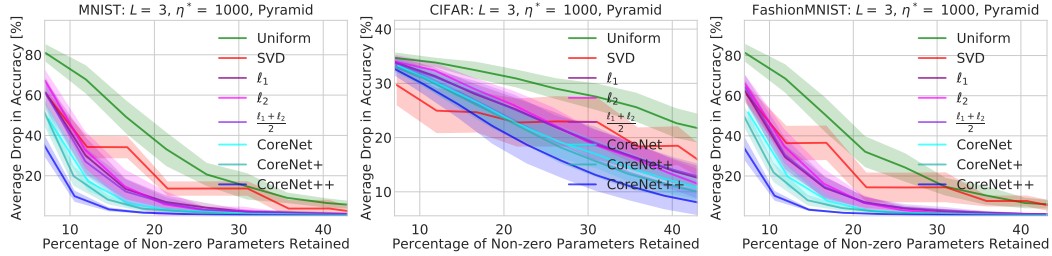

Figure 1: Evaluation of drop in classification accuracy after compression against the *MNIST*, *CIFAR*, and *FashionMNIST* datasets with varying number of hidden layers ($L$) and number of neurons per hidden layer ($\eta^*$). Shaded region corresponds to values within one standard deviation of the mean.

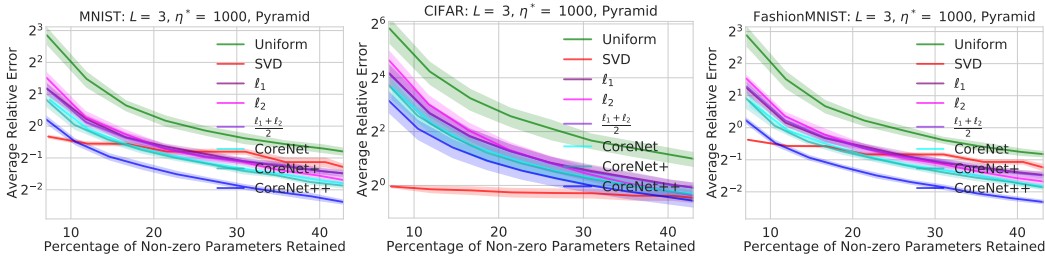

Figure 2: Evaluation of relative error after compression against the *MNIST*, *CIFAR*, and *FashionMNIST* datasets with varying number of hidden layers ($L$) and number of neurons per hidden layer ($\eta^*$).

**Discussion**  The simulation results presented in this section validate our theoretical results established in Sec. 5. In particular, our empirical results indicate that we are able to outperform networks compressed via competing methods in matrix sparsification across all considered experiments and trials. The results presented in this section further suggest that empirical sensitivity can effectively capture the relative importance of neural network parameters, leading to a more informed importance sampling scheme. Moreover, the relative performance of our algorithm tends to increase as we consider deeper architectures. These findings suggest that our algorithm may also be effective in compressing modern convolutional architectures, which tend to be very deep.

## 7  CONCLUSION

We presented a coresets-based neural network compression algorithm for compressing the parameters of a trained fully-connected neural network in a manner that approximately preserves the network's output. Our method and analysis extend traditional coreset constructions to the application of compressing parameters, which may be of independent interest. Our work distinguishes itself from prior approaches in that it establishes theoretical guarantees on the approximation accuracy and size of the generated compressed network. As a corollary to our analysis, we obtain generalization bounds for

neural networks, which may provide novel insights on the generalization properties of neural networks. We empirically demonstrated the practical effectiveness of our compression algorithm on a variety of neural network configurations and real-world data sets. In future work, we plan to extend our algorithm and analysis to compress Convolutional Neural Networks (CNNs) and other network architectures. We conjecture that our compression algorithm can be used to reduce storage requirements of neural network models and enable fast inference in practical settings.

## ACKNOWLEDGMENTS

This research was supported in part by the National Science Foundation award IIS-1723943. We thank Brandon Araki and Kiran Vodrahalli for valuable discussions and helpful suggestions. We would also like to thank Kasper Green Larsen, Alexander Mathiasen, and Allan Gronlund for pointing out an error in an earlier formulation of Lemma 6.

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

## A  PROOFS OF THE ANALYTICAL RESULTS IN SECTION 5

This section includes the full proofs of the technical results given in Sec. 5.

### A.1  ANALYTICAL RESULTS FOR SECTION 5.2 (IMPORTANCE SAMPLING BOUNDS FOR POSITIVE WEIGHTS)

#### A.1.1  ORDER STATISTIC SAMPLING

We now establish a couple of technical results that will quantify the accuracy of our approximations of edge importance (i.e., sensitivity).

**Lemma 6.** *Let $K > 0$ be a universal constant and let $\mathcal{D}$ be a distribution with CDF $F(\cdot)$ satisfying $F(M/K) \leq \exp(-1/K)$, where $M = \min\{x \in [0,1] : F(x) = 1\}$. Let $\mathcal{P} = \{X_1, \ldots, X_n\}$ be a set of $n = |\mathcal{P}|$ i.i.d. samples each drawn from the distribution $\mathcal{D}$. Let $X_{n+1} \sim \mathcal{D}$ be an i.i.d. sample. Then,*

$$\mathbb{P}\left( K \max_{X \in \mathcal{P}} X < X_{n+1} \right) \leq \exp(-n/K)$$

*Proof.* Let $X_{\max} = \max_{X \in \mathcal{P}}$; then,

$$
\begin{aligned}
\mathbb{P}(K\, X_{\max} < X_{n+1}) &= \int_0^M \mathbb{P}(X_{\max} < x/K | X_{n+1} = x)\, d\mathbb{P}(x) \\
&= \int_0^M \mathbb{P}\left(X < x/K\right)^n d\mathbb{P}(x) && \text{since } X_1, \ldots, X_n \text{ are i.i.d.} \\
&\leq \int_0^M F(x/K)^n \, d\mathbb{P}(x) && \text{where } F(\cdot) \text{ is the CDF of } X \sim \mathcal{D} \\
&\leq F(M/K)^n \int_0^M d\mathbb{P}(x) && \text{by monotonicity of } F \\
&= F(M/K)^n \\
&\leq \exp(-n/K) && \text{CDF Assumption,}
\end{aligned}
$$

and this completes the proof.  $\square$

We now proceed to establish that the notion of empirical sensitivity is a good approximation for the relative importance. For this purpose, let the relative importance $\hat{g}_j(x)$ of an edge $j$ after the previous layers have already been compressed be

$$\hat{g}_j(x) = \frac{w_j\, \hat{a}_j(x)}{\sum_{k \in \mathcal{W}} w_k\, \hat{a}_k(x)}.$$

**Lemma 7** (Empirical Sensitivity Approximation). *Let $\varepsilon \in (0, 1/2), \delta \in (0, 1), \ell \in \{2, \ldots, L\}$, Consider a set $\mathcal{S} = \{x_1, \ldots, x_n\} \subseteq \mathcal{P}$ of size $|\mathcal{S}| \geq \lceil \log(8\,\eta\,\eta^*/\delta) \log(\eta\,\eta^*) \rceil$. Then, conditioned on the event $\mathcal{E}_{1/2}$ occurring, i.e., $\hat{a}(x) \in (1 \pm 1/2)a(x)$,*

$$\mathbb{P}_{x \sim \mathcal{D}}\left( \exists j \in \mathcal{W} : C\, s_j < \hat{g}_j(x) \mid \mathcal{E}_{1/2} \right) \leq \frac{\delta}{8\,\eta},$$

*where $C = 3 \log(\eta\,\eta^*)$ and $\mathcal{W} \subseteq [\eta^{\ell-1}]$.*

*Proof.* Consider an arbitrary $j \in \mathcal{W}$ and $x' \in \mathcal{S}$ corresponding to $g_j(x')$ with CDF $F_j(\cdot)$ and recall that $M = \min\{x \in [0,1] : F_j(x) = 1\}$ as in Assumption 1. Note that by Assumption 1, we have

$$F(M/K) \leq \exp(-1/K),$$

and so the random variables $g_j(x')$ for $x' \in \mathcal{S}$ satisfy the CDF condition required by Lemma 6. Now let $\mathcal{E}$ be the event that $K\, s_j < g_j(x)$ holds. Applying Lemma 6, we obtain

$$\mathbb{P}(\mathcal{E}) = \mathbb{P}(K\, s_j < g_j(x)) = \mathbb{P}\left( K \max_{x' \in \mathcal{S}} g_j(x') < g_j(x) \right) \leq \exp(-|\mathcal{S}|/K).$$

Now let $\hat{\mathcal{E}}$ denote the event that the inequality $C s_j < \hat{g}_j(x) = \frac{w_j \, \hat{a}_j(x)}{\sum_{k \in \mathcal{W}} w_k \, \hat{a}_k(x)}$ holds and note that the right side of the inequality is defined with respect to $\hat{g}_j(x)$ and not $g_j(x)$. Observe that since we conditioned on the event $\mathcal{E}_{1/2}$, we have that $\hat{a}(x) \in (1 \pm 1/2)a(x)$.

Now assume that event $\hat{\mathcal{E}}$ holds and note that by the implication above, we have

$$C \, s_j < \hat{g}_j(x) = \frac{w_j \, \hat{a}_j(x)}{\sum_{k \in \mathcal{W}} w_k \, \hat{a}_k(x)} \leq \frac{(1 + 1/2)w_j \, a_j(x)}{(1 - 1/2) \sum_{k \in \mathcal{W}} w_k \, a_k(x)}$$
$$\leq 3 \cdot \frac{w_j \, a_j(x)}{\sum_{k \in \mathcal{W}} w_k \, a_k(x)} = 3 \, g_j(x).$$

where the second inequality follows from the fact that $1+1/2 / 1-1/2 \leq 3$. Moreover, since we know that $C \geq 3K$, we conclude that if event $\hat{\mathcal{E}}$ occurs, we obtain the inequality

$$3 \, K \, s_j \leq 3 \, g_j(x) \Leftrightarrow K \, s_j \leq g_j(x),$$

which is precisely the definition of event $\mathcal{E}$. Thus, we have shown the conditional implication $(\hat{\mathcal{E}} \mid \mathcal{E}_{1/2}) \Rightarrow \mathcal{E}$, which implies that

$$\mathbb{P}(\hat{\mathcal{E}} \mid \mathcal{E}_{1/2}) = \mathbb{P}(C \, s_j < \hat{g}_j(x) \mid \mathcal{E}_{1/2}) \leq \mathbb{P}(\mathcal{E})$$
$$\leq \exp(-|\mathcal{S}|/K).$$

Since our choice of $j \in \mathcal{W}$ was arbitrary, the bound applies for any $j \in \mathcal{W}$. Thus, we have by the union bound

$$\mathbb{P}(\exists j \in \mathcal{W} : C \, s_j < \hat{g}_j(x) \mid \mathcal{E}_{1/2}) \leq \sum_{j \in \mathcal{W}} \mathbb{P}(C \, s_j < \hat{g}_j(x) \mid \mathcal{E}_{1/2}) \leq |\mathcal{W}| \exp(-|\mathcal{S}|/K)$$
$$= \left( \frac{|\mathcal{W}|}{\eta^*} \right) \frac{\delta}{8\eta} \leq \frac{\delta}{8\eta}.$$

$\square$

In practice, the set $\mathcal{S}$ referenced above is chosen to be a subset of the original data points, i.e., $\mathcal{S} \subseteq \mathcal{P}$ (see Alg. 1, Line 2). Thus, we henceforth assume that the size of the input points $|\mathcal{P}|$ is large enough (or the specified parameter $\delta \in (0, 1)$ is sufficiently large) so that $|\mathcal{P}| \geq |\mathcal{S}|$.

### A.1.2 Proof of Lemma 1

We now state the proof of Lemma 1. In this subsection, we establish approximation guarantees under the assumption that the weights are strictly positive. The next subsection will then relax this assumption to conclude that a neuron's value can be approximated well even when the weights are not all positive.

**Lemma 1** (Positive-Weights Sparsification). *Let* $\varepsilon, \delta \in (0, 1)$, *and* $x \sim \mathcal{D}$. SPARSIFY$(\mathcal{W}, w, \varepsilon, \delta, \mathcal{S}, a(\cdot))$ *generates a row vector* $\hat{w}$ *such that*

$$\mathbb{P}\left( \sum_{k \in \mathcal{W}} \hat{w}_k \, \hat{a}_k(x) \notin (1 \pm \varepsilon) \sum_{k \in \mathcal{W}} w_k \, \hat{a}_k(x) \mid \mathcal{E}_{1/2} \right) \leq \frac{3\delta}{8\eta}$$

*where* $\mathrm{nnz}(\hat{w}) \leq \left\lceil \frac{8 \, S \, \log(\eta \, \eta^*) \log(8 \, \eta/\delta)}{\varepsilon^2} \right\rceil$, *and* $S = \sum_{j \in \mathcal{W}} s_j$.

*Proof.* Let $\varepsilon, \delta \in (0, 1)$ be arbitrary. Moreover, let $\mathcal{C}$ be the coreset with respect to the weight indices $\mathcal{W} \subseteq [\eta^{\ell-1}]$ used to construct $\hat{w}$. Note that as in SPARSIFY, $\mathcal{C}$ is a multiset sampled from $\mathcal{W}$ of size $m = \left\lceil \frac{8 \, S \, \log(\eta \, \eta^*) \log(8 \, \eta/\delta)}{\varepsilon^2} \right\rceil$, where $S = \sum_{j \in \mathcal{W}} s_j$ and $\mathcal{C}$ is sampled according to the probability distribution $q$ defined by

$$q_j = \frac{s_j}{S} \qquad \forall j \in \mathcal{W}.$$

Let $\hat{\mathbf{a}}(\cdot)$ be an arbitrary realization of the random variable $\hat{a}^{\ell-1}(\cdot)$, let $\mathbf{x}$ be a realization of $x \sim \mathcal{D}$, and let

$$\hat{z} = \sum_{k \in \mathcal{W}} \hat{w}_k \, \hat{\mathbf{a}}_k(\mathbf{x})$$

be the approximate intermediate value corresponding to the sparsified matrix $\hat{w}$ and let

$$\tilde{z} = \sum_{k \in \mathcal{W}} w_k \, \hat{\mathbf{a}}_k(\mathbf{x}).$$

Now define $\mathcal{E}$ to be the (favorable) event that $\hat{z}$ $\varepsilon$-approximates $\tilde{z}$, i.e., $\hat{z} \in (1 \pm \varepsilon)\tilde{z}$, We will now show that the complement of this event, $\mathcal{E}^c$, occurs with sufficiently small probability. Let $\mathcal{Z} \subseteq \operatorname{supp}(\mathcal{D})$ be the set of *well-behaved* points (defined implicitly with respect to neuron $i \in [\eta^\ell]$ and realization $\hat{\mathbf{a}}$) and defined as follows:

$$\mathcal{Z} = \{x' \in \operatorname{supp}(\mathcal{D}) \,:\, \hat{g}_j(x') \leq C s_j \quad \forall j \in \mathcal{W}\},$$

where $C = 3 \log(\eta \, \eta^*)$. Let $\mathcal{E}_{\mathcal{Z}}$ denote the event that $\mathbf{x} \in \mathcal{Z}$ where $\mathbf{x}$ is a realization of $x \sim \mathcal{D}$.

**Conditioned on $\mathcal{E}_{\mathcal{Z}}$, event $\mathcal{E}^c$ occurs with probability $\leq \frac{\delta}{4\eta}$:** Let $\mathbf{x}$ be a realization of $x \sim \mathcal{D}$ such that $\mathbf{x} \in \mathcal{Z}$ and let $\mathcal{C} = \{c_1, \ldots, c_m\}$ be $m$ samples from $\mathcal{W}$ with respect to distribution $q$ as before. Define $m$ random variables $T_{c_1}, \ldots, T_{c_m}$ such that for all $j \in \mathcal{C}$

$$T_j = \frac{w_j \, \hat{\mathbf{a}}_j(\mathbf{x})}{m \, q_j} = \frac{S \, w_j \, \hat{\mathbf{a}}_j(\mathbf{x})}{m \, s_j}. \tag{2}$$

For any $j \in \mathcal{C}$, we have for the conditional expectation of $T_j$:

$$\mathbb{E}\left[T_j \mid \hat{\mathbf{a}}(\cdot), \mathbf{x}, \mathcal{E}_{\mathcal{Z}}, \mathcal{E}_{1/2}\right] = \sum_{k \in \mathcal{W}} \frac{w_k \, \hat{\mathbf{a}}_k(\mathbf{x})}{m \, q_k} \cdot q_k$$

$$= \sum_{k \in \mathcal{W}} \frac{w_k \, \hat{\mathbf{a}}_k(\mathbf{x})}{m}$$

$$= \frac{\tilde{z}}{m},$$

where we use the expectation notation $\mathbb{E}[\cdot]$ with the understanding that it denotes the conditional expectation $\mathbb{E}_{\mathcal{C} \mid \hat{a}^{l-1}(\cdot), x}[\cdot]$. Moreover, we also note that conditioning on the event $\mathcal{E}_{\mathcal{Z}}$ (i.e., the event that $\mathbf{x} \in \mathcal{Z}$) does not affect the expectation of $T_j$. Let $T = \sum_{j \in \mathcal{C}} T_j = \hat{z}$ denote our approximation and note that by linearity of expectation,

$$\mathbb{E}\left[T \mid \hat{\mathbf{a}}(\cdot), \mathbf{x}, \mathcal{E}_{\mathcal{Z}}, \mathcal{E}_{1/2}\right] = \sum_{j \in \mathcal{C}} \mathbb{E}\left[T_j \mid \hat{\mathbf{a}}(\cdot), \mathbf{x}, \mathcal{E}_{\mathcal{Z}}, \mathcal{E}_{1/2}\right] = \tilde{z}$$

Thus, $\hat{z} = T$ is an unbiased estimator of $\tilde{z}$ for any realization $\hat{\mathbf{a}}(\cdot)$ and $\mathbf{x}$; thus, we will henceforth refer to $\mathbb{E}[T \mid \hat{\mathbf{a}}(\cdot), \mathbf{x}]$ as simply $\tilde{z}$ for brevity.

For the remainder of the proof we will assume that $\tilde{z} > 0$, since otherwise, $\tilde{z} = 0$ if and only if $T_j = 0$ for all $j \in \mathcal{C}$ almost surely, which follows by the fact that $T_j \geq 0$ for all $j \in \mathcal{C}$ by definition of $\mathcal{W}$ and the non-negativity of the ReLU activation. Therefore, in the case that $\tilde{z} = 0$, it follows that

$$\mathbb{P}(|\hat{z} - \tilde{z}| > \varepsilon\tilde{z} \mid \hat{\mathbf{a}}(\cdot), \mathbf{x}) = \mathbb{P}(\hat{z} > 0 \mid \hat{\mathbf{a}}(\cdot), \mathbf{x}) = \mathbb{P}(0 > 0) = 0,$$

which trivially yields the statement of the lemma, where in the above expression, $\mathbb{P}(\cdot)$ is short-hand for the conditional probability $\mathbb{P}_{\hat{w} \mid \hat{a}^{l-1}(\cdot), x}(\cdot)$.

We now proceed with the case where $\tilde{z} > 0$ and leverage the fact that $\mathbf{x} \in \mathcal{Z}$[3] to establish that for all $j \in \mathcal{W}$:

$$C s_j \geq \hat{g}_j(\mathbf{x}) = \frac{w_j \, \hat{\mathbf{a}}_j(\mathbf{x})}{\sum_{k \in \mathcal{W}} w_k \, \hat{\mathbf{a}}_k(\mathbf{x})} = \frac{w_j \, \hat{\mathbf{a}}_j(\mathbf{x})}{\tilde{z}}$$

---

[3]Since we conditioned on the event $\mathcal{E}_{\mathcal{Z}}$.

$$\Leftrightarrow \quad \frac{w_j\,\hat{\mathbf{a}}_j(\mathbf{x})}{s_j} \le C\,\tilde{z}. \tag{3}$$

Utilizing the inequality established above, we bound the conditional variance of each $T_j$, $j \in \mathcal{C}$ as follows

$$
\begin{aligned}
\mathrm{Var}(T_j \mid \hat{\mathbf{a}}(\cdot), \mathbf{x}, \mathcal{E}_\mathcal{Z}, \mathcal{E}_{1/2}) &\le \mathbb{E}\left[(T_j)^2 \mid \hat{\mathbf{a}}(\cdot), \mathbf{x}, \mathcal{E}_\mathcal{Z}, \mathcal{E}_{1/2}\right] \\
&= \sum_{k \in \mathcal{W}} \frac{(w_k\,\hat{\mathbf{a}}_k(\mathbf{x}))^2}{(m\,q_k)^2} \cdot q_k \\
&= \frac{S}{m^2} \sum_{k \in \mathcal{W}} \frac{(w_k\,\hat{\mathbf{a}}_k(\mathbf{x}))^2}{s_k} \\
&\le \frac{S}{m^2} \left(\sum_{k \in \mathcal{W}} w_k\,\hat{\mathbf{a}}_k(\mathbf{x})\right) C\,\tilde{z} \\
&= \frac{S\,C\,\tilde{z}^2}{m^2},
\end{aligned}
$$

where $\mathrm{Var}(\cdot)$ is short-hand for $\mathrm{Var}_{\mathcal{C}\,|\,\hat{a}^{l-1}(\cdot),\,x}(\cdot)$. Since $T$ is a sum of (conditionally) independent random variables, we obtain

$$
\begin{aligned}
\mathrm{Var}(T \mid \hat{\mathbf{a}}(\cdot), \mathbf{x}, \mathcal{E}_\mathcal{Z}, \mathcal{E}_{1/2}) &= m\,\mathrm{Var}(T_j \mid \hat{\mathbf{a}}(\cdot), \mathbf{x}, \mathcal{E}_\mathcal{Z}, \mathcal{E}_{1/2}) \tag{4} \\
&\le \frac{S\,C\,\tilde{z}^2}{m}.
\end{aligned}
$$

Now, for each $j \in \mathcal{C}$ let

$$\widetilde{T}_j = T_j - \mathbb{E}\left[T_j \mid \hat{\mathbf{a}}(\cdot), \mathbf{x}, \mathcal{E}_\mathcal{Z}, \mathcal{E}_{1/2}\right] = T_j - \tilde{z},$$

and let $\widetilde{T} = \sum_{j \in \mathcal{C}} \widetilde{T}_j$. Note that by the fact that we conditioned on the realization $\mathbf{x}$ of $x$ such that $\mathbf{x} \in \mathcal{Z}$ (event $\mathcal{E}_\mathcal{Z}$), we obtain by definition of $T_j$ in (2) and the inequality (3):

$$T_j = \frac{S\,w_j\,\hat{\mathbf{a}}_j(\mathbf{x})}{m\,s_j} \le \frac{S\,C\,\tilde{z}}{m}. \tag{5}$$

We also have that $S \ge 1$ by definition. More specifically, using the fact that the maximum over a set is greater than the average and rearranging sums, we obtain

$$
\begin{aligned}
S = \sum_{j \in \mathcal{W}} s_j &= \sum_{j \in \mathcal{W}} \max_{\mathbf{x}' \in \mathcal{S}} g_j(\mathbf{x}') \\
&\ge \frac{1}{|\mathcal{S}|} \sum_{j \in \mathcal{W}} \sum_{\mathbf{x}' \in \mathcal{S}} g_j(\mathbf{x}') = \frac{1}{|\mathcal{S}|} \sum_{\mathbf{x}' \in \mathcal{S}} \sum_{j \in \mathcal{W}} g_j(\mathbf{x}') \\
&= \frac{1}{|\mathcal{S}|} \sum_{\mathbf{x}' \in \mathcal{S}} 1 = 1.
\end{aligned}
$$

Thus, the inequality established in (5) with the fact that $S \ge 1$ we obtain an upper bound on the absolute value of the centered random variables:

$$|\widetilde{T}_j| = \left|T_j - \frac{\tilde{z}}{m}\right| \le \frac{S\,C\,\tilde{z}}{m} = M, \tag{6}$$

which follows from the fact that: **if** $T_j \ge \frac{\tilde{z}}{m}$**:** Then, by our bound in (5) and the fact that $\frac{\tilde{z}}{m} \ge 0$, it follows that

$$\left|\widetilde{T}_j\right| = T_j - \frac{\tilde{z}}{m} \le \frac{S\,C\,\tilde{z}}{m} - \frac{\tilde{z}}{m} \le \frac{S\,C\,\tilde{z}}{m}.$$

**if $T_j < \frac{\tilde{z}}{m}$:** Then, using the fact that $T_j \geq 0$ and $S \geq 1$, we obtain

$$\left|\widetilde{T}_j\right| = \frac{\tilde{z}}{m} - T_j \leq \frac{\tilde{z}}{m} \leq \frac{S\,C\,\tilde{z}}{m}.$$

Applying Bernstein's inequality to both $\widetilde{T}$ and $-\widetilde{T}$ we have by symmetry and the union bound,

$$\mathbb{P}(\mathcal{E}^{\mathsf{c}} \mid \hat{\mathbf{a}}(\cdot), \mathbf{x}, \mathcal{E}_{\mathcal{Z}}, \mathcal{E}_{1/2}) = \mathbb{P}\left(|T - \tilde{z}| \geq \varepsilon\tilde{z} \mid \hat{\mathbf{a}}(\cdot), \mathbf{x}, \mathcal{E}_{\mathcal{Z}}, \mathcal{E}_{1/2}\right)$$

$$\leq 2\exp\left(-\frac{\varepsilon^2\tilde{z}^2}{2\operatorname{Var}(T \mid \hat{\mathbf{a}}(\cdot), \mathbf{x}) + \frac{2\,\varepsilon\,\tilde{z}M}{3}}\right)$$

$$\leq 2\exp\left(-\frac{\varepsilon^2\tilde{z}^2}{\frac{2SC\,\tilde{z}^2}{m} + \frac{2S\,C\,\tilde{z}^2}{3m}}\right)$$

$$= 2\exp\left(-\frac{3\,\varepsilon^2\,m}{8S\,C}\right)$$

$$\leq \frac{\delta}{4\eta},$$

where the second inequality follows by our upper bounds on $\operatorname{Var}(T \mid \hat{\mathbf{a}}(\cdot), \mathbf{x})$ and $\left|\widetilde{T}_j\right|$ and the fact that $\varepsilon \in (0, 1)$, and the last inequality follows by our choice of $m = \left\lceil \frac{8\,S\,\log(\eta\,\eta^*)\,\log(8\,\eta/\delta)}{\varepsilon^2} \right\rceil$. This establishes that for any realization $\hat{\mathbf{a}}(\cdot)$ of $\hat{a}^{l-1}(\cdot)$ and a realization $\mathbf{x}$ of $x$ satisfying $\mathbf{x} \in \mathcal{Z}$, the event $\mathcal{E}^{\mathsf{c}}$ occurs with probability at most $\frac{\delta}{4\eta}$.

**Removing the conditioning on $\mathcal{E}_{\mathcal{Z}}$:** We have by law of total probability

$$\mathbb{P}(\mathcal{E} \mid \hat{\mathbf{a}}(\cdot), \mathcal{E}_{1/2}) \geq \int_{\mathbf{x} \in \mathcal{Z}} \mathbb{P}(\mathcal{E} \mid \hat{\mathbf{a}}(\cdot), \mathbf{x}, \mathcal{E}_{\mathcal{Z}}, \mathcal{E}_{1/2}) \, \underset{x \sim \mathcal{D}}{\mathbb{P}}(x = \mathbf{x} \mid \hat{\mathbf{a}}(\cdot), \mathcal{E}_{1/2}) \, d\mathbf{x}$$

$$\geq \left(1 - \frac{\delta}{4\eta}\right) \int_{\mathbf{x} \in \mathcal{Z}} \underset{x \sim \mathcal{D}}{\mathbb{P}}(x = \mathbf{x} \mid \hat{\mathbf{a}}(\cdot), \mathcal{E}_{1/2}) \, d\mathbf{x}$$

$$= \left(1 - \frac{\delta}{4\eta}\right) \underset{x \sim \mathcal{D}}{\mathbb{P}}(\mathcal{E}_{\mathcal{Z}} \mid \hat{\mathbf{a}}(\cdot), \mathcal{E}_{1/2})$$

$$\geq \left(1 - \frac{\delta}{4\eta}\right)\left(1 - \frac{\delta}{8\eta}\right)$$

$$\geq 1 - \frac{3\delta}{8\eta}$$

where the second-to-last inequality follows from the fact that $\mathbb{P}(\mathcal{E}^{\mathsf{c}} \mid \hat{\mathbf{a}}(\cdot), \mathbf{x}, \mathcal{E}_{\mathcal{Z}}, \mathcal{E}_{1/2}) \leq \frac{\delta}{4\eta}$ as was established above and the last inequality follows by Lemma 7.

**Putting it all together** Finally, we marginalize out the random variable $\hat{a}^{\ell-1}(\cdot)$ to establish

$$\mathbb{P}(\mathcal{E} \mid \mathcal{E}_{1/2}) = \int_{\hat{\mathbf{a}}(\cdot)} \mathbb{P}(\mathcal{E} \mid \hat{\mathbf{a}}(\cdot), \mathcal{E}_{1/2})\, \mathbb{P}(\hat{\mathbf{a}}(\cdot) \mid \mathcal{E}_{1/2}) \, d\hat{\mathbf{a}}(\cdot)$$

$$\geq \left(1 - \frac{3\delta}{8\eta}\right) \int_{\hat{\mathbf{a}}(\cdot)} \mathbb{P}(\hat{\mathbf{a}}(\cdot) \mid \mathcal{E}_{1/2}) \, d\hat{\mathbf{a}}(\cdot)$$

$$= 1 - \frac{3\delta}{8\eta}.$$

Consequently,

$$\mathbb{P}(\mathcal{E}^{\mathsf{c}} \mid \mathcal{E}_{1/2}) \leq 1 - \left(1 - \frac{3\delta}{8\eta}\right) = \frac{3\delta}{8\eta},$$

and this concludes the proof. $\qquad\square$

## A.2 Analytical Results for Section 5.3 (Importance Sampling Bounds)

We begin by establishing an auxiliary result that we will need for the subsequent lemmas.

### A.2.1 Empirical $\Delta_i^\ell$ Approximation

**Lemma 8** (Empirical $\Delta_i^\ell$ Approximation). *Let* $\delta \in (0,1)$, $\lambda_* = \log(\eta \, \eta^*)/2$, *and define*

$$\hat{\Delta}^\ell = \left( \frac{1}{|\mathcal{S}|} \max_{i \in [\eta^\ell]} \sum_{x' \in \mathcal{S}} \Delta_i^\ell(x') \right) + \kappa,$$

*where* $\kappa = \sqrt{2\lambda_*} \left( 1 + \sqrt{2\lambda_*} \log \left( 8 \, \eta \, \eta^* / \delta \right) \right)$ *and* $\mathcal{S} \subseteq \mathcal{P}$ *is as in Alg. 1. Then,*

$$\mathbb{P}_{x \sim \mathcal{D}} \left( \max_{i \in [\eta^\ell]} \Delta_i^\ell(x) \leq \hat{\Delta}^\ell \right) \geq 1 - \frac{\delta}{4\eta}.$$

*Proof.* Define the random variables $\mathcal{Y}_{x'} = \mathbb{E}\left[\Delta_i^\ell(x')\right] - \Delta_i^\ell(x')$ for each $x' \in \mathcal{S}$ and consider the sum

$$\mathcal{Y} = \sum_{x' \in \mathcal{S}} \mathcal{Y}_{x'} = \sum_{x' \in \mathcal{S}} \left( \mathbb{E}\left[\Delta_i^\ell(x)\right] - \Delta_i^\ell(x') \right).$$

We know that each random variable $\mathcal{Y}_{\mathbf{x}'}$ satisfies $\mathbb{E}\left[\mathcal{Y}_{\mathbf{x}'}\right] = 0$ and by Assumption 2, is subexponential with parameter $\lambda \leq \lambda_*$. Thus, $\mathcal{Y}$ is a sum of $|\mathcal{S}|$ independent, zero-mean $\lambda_*$-subexponential random variables, which implies that $\mathbb{E}\left[\mathcal{Y}\right] = 0$ and that we can readily apply Bernstein's inequality for subexponential random variables (Vershynin, 2016) to obtain for $t \geq 0$

$$\mathbb{P}\left( \frac{1}{|\mathcal{S}|} \mathcal{Y} \geq t \right) \leq \exp\left( -|\mathcal{S}| \min\left\{ \frac{t^2}{4\lambda_*^2}, \frac{t}{2\lambda_*} \right\} \right).$$

Since $\mathcal{S} = \lceil \log \left( 8 \, \eta \, \eta^* / \delta \right) \log(\eta \, \eta^*) \rceil \geq \log \left( 8 \, \eta \, \eta^* / \delta \right) 2\lambda^*$, we have for $t = \sqrt{2\lambda_*}$,

$$\mathbb{P}\left( \mathbb{E}\left[\Delta_i^\ell(x)\right] - \frac{1}{|\mathcal{S}|} \sum_{x' \in \mathcal{S}} \Delta_i^\ell(x') \geq t \right) = \mathbb{P}\left( \frac{1}{|\mathcal{S}|} \mathcal{Y} \geq t \right)$$

$$\leq \exp\left( -|\mathcal{S}| \frac{t^2}{4\lambda_*^2} \right)$$

$$\leq \exp\left( -\log \left( 8 \, \eta \, \eta^* / \delta \right) \right)$$

$$= \frac{\delta}{8 \, \eta \, \eta^*}.$$

Moreover, for a single $\mathcal{Y}_x$, we have by the equivalent definition of a subexponential random variable (Vershynin, 2016) that for $u \geq 0$

$$\mathbb{P}(\Delta_i^\ell(x) - \mathbb{E}\left[\Delta_i^\ell(x)\right] \geq u) \leq \exp\left( -\min\left\{ \frac{u^2}{4\lambda_*^2}, \frac{u}{2\lambda_*} \right\} \right).$$

Thus, for $u = 2\lambda_* \log \left( 8 \, \eta \, \eta^* / \delta \right)$ we obtain

$$\mathbb{P}(\Delta_i^\ell(x) - \mathbb{E}\left[\Delta_i^\ell(x)\right] \geq u) \leq \exp\left( -\log \left( 8 \, \eta \, \eta^* / \delta \right) \right) = \frac{\delta}{8 \, \eta \, \eta^*}.$$

Therefore, by the union bound, we have with probability at least $1 - \frac{\delta}{4\eta \, \eta^*}$:

$$\Delta_i^\ell(x) \leq \mathbb{E}\left[\Delta_i^\ell(x)\right] + u$$

$$\leq \left( \frac{1}{|\mathcal{S}|} \sum_{\mathbf{x}' \in \mathcal{S}} \Delta_i^\ell(x') + t \right) + u$$

$$= \frac{1}{|\mathcal{S}|} \sum_{x' \in \mathcal{S}} \Delta_i^\ell(x') + \left( \sqrt{2\lambda_*} + 2\lambda_* \log \left( 8 \, \eta \, \eta^* / \delta \right) \right)$$

$$= \frac{1}{|\mathcal{S}|} \sum_{x' \in \mathcal{S}} \Delta_i^\ell(x') + \kappa$$

$$\leq \hat{\Delta}^\ell,$$

where the last inequality follows by definition of $\hat{\Delta}^\ell$.

Thus, by the union bound, we have

$$\mathbb{P}_{x \sim \mathcal{D}} \left( \max_{i \in [\eta^\ell]} \Delta_i^\ell(x) > \hat{\Delta}^\ell \right) = \mathbb{P} \left( \exists i \in [\eta^\ell] : \Delta_i^\ell(x) > \hat{\Delta}^\ell \right)$$

$$\leq \sum_{i \in [\eta^\ell]} \mathbb{P} \left( \Delta_i^\ell(x) > \hat{\Delta}^\ell \right)$$

$$\leq \eta^\ell \left( \frac{\delta}{4\eta \, \eta^*} \right)$$

$$\leq \frac{\delta}{4 \, \eta},$$

where the last line follows by definition of $\eta^* \geq \eta^\ell$. $\qquad\square$

### A.2.2 Notation for the Subsequent Analysis

Let $\hat{w}_i^{\ell+}$ and $\hat{w}_i^{\ell-}$ denote the sparsified row vectors generated when SPARSIFY is invoked with first two arguments corresponding to $(\mathcal{W}_+, w_i^\ell)$ and $(\mathcal{W}_-, -w_i^\ell)$, respectively (Alg. 1, Line 12). We will at times omit including the variables for the neuron $i$ and layer $\ell$ in the proofs for clarity of exposition, and for example, refer to $\hat{w}_i^{\ell+}$ and $\hat{w}_i^{\ell-}$ as simply $\hat{w}^+$ and $\hat{w}^-$, respectively.

Let $x \sim \mathcal{D}$ and define

$$\hat{z}^+(x) = \sum_{k \in \mathcal{W}_+} \hat{w}_k^+ \, \hat{a}_k(x) \geq 0 \qquad \text{and} \qquad \hat{z}^-(x) = \sum_{k \in \mathcal{W}_-} (-\hat{w}_k^-) \, \hat{a}_k(x) \geq 0$$

be the approximate intermediate values corresponding to the sparsified matrices $\hat{w}^+$ and $\hat{w}^-$; let

$$\tilde{z}^+(x) = \sum_{k \in \mathcal{W}_+} w_k \, \hat{a}_k(x) \geq 0 \qquad \text{and} \qquad \tilde{z}^-(x) = \sum_{k \in \mathcal{W}_-} (-w_k) \, \hat{a}_k(x) \geq 0$$

be the corresponding intermediate values with respect to the the original row vector $w$; and finally, let

$$z^+(x) = \sum_{k \in \mathcal{W}_+} w_k \, a_k(x) \geq 0 \qquad \text{and} \qquad z^-(x) = \sum_{k \in \mathcal{W}_-} (-w_k) \, a_k(x) \geq 0$$

be the true intermediate values corresponding to the positive and negative valued weights.

Note that in this context, we have by definition

$$\hat{z}_i^\ell(x) = \langle \hat{w}, \hat{a}(x) \rangle = \hat{z}^+(x) - \hat{z}^-(x),$$

$$\tilde{z}_i^\ell(x) = \langle w, \hat{a}(x) \rangle = \tilde{z}^+(x) - \tilde{z}^-(x), \quad \text{and}$$

$$z_i^\ell(x) = \langle w, a(x) \rangle = z^+(x) - z^-(x),$$

where we used the fact that $\hat{w} = \hat{w}^+ - \hat{w}^- \in \mathbb{R}^{1 \times \eta^{\ell-1}}$.

### A.2.3 Proof of Lemma 2

**Lemma 2** (Conditional Neuron Value Approximation). *Let $\varepsilon, \delta \in (0,1)$, $\ell \in \{2, \ldots, L\}$, $i \in [\eta^\ell]$, and $x \sim \mathcal{D}$. CORENET generates a row vector $\hat{w}_i^\ell = \hat{w}_i^{\ell+} - \hat{w}_i^{\ell-} \in \mathbb{R}^{1 \times \eta^{\ell-1}}$ such that*

$$\mathbb{P} \left( \mathcal{E}_i^\ell \mid \mathcal{E}^{\ell-1} \right) = \mathbb{P} \left( \hat{z}_i^\ell(x) \in (1 \pm 2\,(\ell-1)\,\varepsilon_{\ell+1})\, z_i^\ell(x) \mid \mathcal{E}^{\ell-1} \right) \geq 1 - \delta/\eta, \tag{1}$$

*where $\varepsilon_\ell = \frac{\varepsilon'}{\hat{\Delta}^{\ell \to}}$ and $\mathrm{nnz}(\hat{w}_i^\ell) \leq \left\lceil \frac{8\,S\,\log(\eta\,\eta^*)\,\log(8\,\eta/\delta)}{\varepsilon_\ell^2} \right\rceil + 1$, where $S = \sum_{j \in \mathcal{W}_+} s_j + \sum_{j \in \mathcal{W}_-} s_j$.*

*Proof.* Let $\varepsilon, \delta \in (0,1)$ be arbitrary and let $\mathcal{W}_+ = \{j \in [\eta^{\ell-1}] : w_j > 0\}$ and $\mathcal{W}_- = \{j \in [\eta^{\ell-1}] : w_j < 0\}$ as in Alg. 1. Let $\varepsilon_\ell$ be defined as before, $\varepsilon_\ell = \frac{\varepsilon'}{\hat{\Delta}^{\ell\to}}$, where $\hat{\Delta}^{\ell\to} = \prod_{k=\ell}^{L} \hat{\Delta}^k$ and $\hat{\Delta}^\ell = \left( \frac{1}{|\mathcal{S}|} \max_{i \in [\eta^\ell]} \sum_{x' \in \mathcal{S}} \Delta_i^\ell(x') \right) + \kappa$.

Observe that $w_j > 0 \;\forall j \in \mathcal{W}_+$ and similarly, for all $(-w_j) > 0 \;\forall j \in \mathcal{W}_-$. That is, each of index sets $\mathcal{W}_+$ and $\mathcal{W}_-$ corresponds to strictly positive entries in the arguments $w_i^\ell$ and $-w_i^\ell$, respectively passed into SPARSIFY. Observe that since we conditioned on the event $\mathcal{E}^{\ell-1}$, we have

$$
\begin{aligned}
2(\ell-2)\varepsilon_\ell &\leq 2(\ell-2) \frac{\varepsilon}{2(L-1) \prod_{k=\ell}^{L} \hat{\Delta}^k} \\
&\leq \frac{\varepsilon}{\prod_{k=\ell}^{L} \hat{\Delta}^k} \\
&\leq \frac{\varepsilon}{2^{L-\ell+1}} \qquad\qquad \text{Since } \hat{\Delta}^k \geq 2 \quad \forall k \in \{\ell, \ldots, L\} \\
&\leq \frac{\varepsilon}{2},
\end{aligned}
$$

where the inequality $\hat{\Delta}^k \geq 2$ follows from the fact that

$$
\begin{aligned}
\hat{\Delta}^k &= \left( \frac{1}{|\mathcal{S}|} \max_{i \in [\eta^\ell]} \sum_{\mathbf{x}' \in \mathcal{S}} \Delta_i^\ell(\mathbf{x}') \right) + \kappa \\
&\geq 1 + \kappa \qquad\qquad \text{Since } \Delta_i^\ell(\mathbf{x}') \geq 1 \quad \forall \mathbf{x}' \in \mathrm{supp}(\mathcal{D}) \text{ by definition} \\
&\geq 2.
\end{aligned}
$$

we obtain that $\hat{a}(x) \in (1 \pm \varepsilon/2) a(x)$, where, as before, $\hat{a}$ and $a$ are shorthand notations for $\hat{a}^{\ell-1} \in \mathbb{R}^{\eta^{\ell-1} \times 1}$ and $a^{\ell-1} \in \mathbb{R}^{\eta^{\ell-1} \times 1}$, respectively. This implies that $\mathcal{E}^{\ell-1} \Rightarrow \mathcal{E}_{1/2}$ and since $m = \left\lceil \frac{8 \, S \, \log(\eta \, \eta^*) \log(8 \, \eta/\delta)}{\varepsilon^2} \right\rceil$ in Alg. 2 we can invoke Lemma 1 with $\varepsilon = \varepsilon_\ell$ on each of the SPARSIFY invocations to conclude that

$$
\mathbb{P}\left( \hat{z}^+(x) \notin (1 \pm \varepsilon_\ell) \tilde{z}^+(x) \mid \mathcal{E}^{\ell-1} \right) \leq \mathbb{P}\left( \hat{z}^+(x) \notin (1 \pm \varepsilon_\ell) \tilde{z}^+(x) \mid \mathcal{E}_{1/2} \right) \leq \frac{3\delta}{8\eta},
$$

and

$$
\mathbb{P}\left( \hat{z}^-(x) \notin (1 \pm \varepsilon_\ell) \tilde{z}^-(x) \mid \mathcal{E}^{\ell-1} \right) \leq \frac{3\delta}{8\eta}.
$$

Therefore, by the union bound, we have

$$
\mathbb{P}\left( \hat{z}^+(x) \notin (1 \pm \varepsilon_\ell) \tilde{z}^+(x) \text{ or } \hat{z}^-(x) \notin (1 \pm \varepsilon_\ell) \tilde{z}^-(x) \mid \mathcal{E}^{\ell-1} \right) \leq \frac{3\delta}{8\eta} + \frac{3\delta}{8\eta} = \frac{3\delta}{4\eta}.
$$

Moreover, by Lemma 8, we have with probability at most $\frac{\delta}{4\eta}$ that

$$
\Delta_i^\ell(x) > \hat{\Delta}^\ell.
$$

Thus, by the union bound over the failure events, we have that with probability at least $1 - (3\delta/4\eta + \delta/4\eta) = 1 - \delta/\eta$ that **both** of the following events occur

1. $\hat{z}^+(x) \in (1 \pm \varepsilon_\ell) \tilde{z}^+(x)$ and $\hat{z}^-(x) \in (1 \pm \varepsilon_\ell) \tilde{z}^-(x)$       (7)

2. $\Delta_i^\ell(x) \leq \hat{\Delta}^\ell$       (8)

Recall that $\varepsilon' = \frac{\varepsilon}{2(L-1)}$, $\varepsilon_\ell = \frac{\varepsilon'}{\hat{\Delta}^{\ell\to}}$, and that event $\mathcal{E}_i^\ell$ denotes the (desirable) event that

$$
\hat{z}_i^\ell(x) \left( 1 \pm 2(\ell-1)\varepsilon_{\ell+1} \right) z_i^\ell(x)
$$

holds, and similarly, $\mathcal{E}^\ell = \cap_{i \in [\eta^\ell]} \mathcal{E}_i^\ell$ denotes the vector-wise analogue where

$$
\hat{z}^\ell(x) \left( 1 \pm 2(\ell-1)\varepsilon_{\ell+1} \right) z^\ell(x).
$$

Let $k = 2\,(\ell - 1)$ and note that by conditioning on the event $\mathcal{E}^{\ell-1}$, i.e., we have by definition

$$\hat{a}^{\ell-1}(x) \in (1 \pm 2\,(\ell - 2)\varepsilon_\ell)a^{\ell-1}(x) = (1 \pm k\,\varepsilon_\ell)a^{\ell-1}(x),$$

which follows by definition of the ReLU function. Recall that our overarching goal is to establish that

$$\hat{z}_i^\ell(x) \in (1 \pm 2\,(\ell - 1)\varepsilon_{\ell+1})\,z_i^\ell(x),$$

which would immediately imply by definition of the ReLU function that

$$\hat{a}_i^\ell(x) \in (1 \pm 2\,(\ell - 1)\varepsilon_{\ell+1})\,a_i^\ell(x).$$

Having clarified the conditioning and our objective, we will once again drop the index $i$ from the expressions moving forward.

Proceeding from above, we have with probability at least $1 - \delta/\eta$

$$
\begin{aligned}
\hat{z}(x) = \hat{z}^+(x) - \hat{z}^-(x) \\
\leq (1 + \varepsilon_\ell)\,\tilde{z}^+(x) - (1 - \varepsilon_\ell)\,\tilde{z}^-(x) &\qquad \text{By Event (7) above} \\
\leq (1 + \varepsilon_\ell)(1 + k\,\varepsilon_\ell)\,z^+(x) - (1 - \varepsilon_\ell)(1 - k\,\varepsilon_\ell)\,z^-(x) &\qquad \text{Conditioning on event } \mathcal{E}^{\ell-1} \\
= \left(1 + \varepsilon_\ell(k + 1) + k\varepsilon_\ell^2\right)z^+(x) + \left(-1 + (k + 1)\varepsilon_\ell - k\varepsilon_\ell^2\right)z^-(x) \\
= \left(1 + k\,\varepsilon_\ell^2\right)z(x) + (k + 1)\,\varepsilon_\ell\left(z^+(x) + z^-(x)\right) \\
= \left(1 + k\,\varepsilon_\ell^2\right)z(x) + \frac{(k + 1)\,\varepsilon'}{\prod_{k=\ell}^{L}\hat{\Delta}^k}\left(z^+(x) + z^-(x)\right) \\
\leq \left(1 + k\,\varepsilon_\ell^2\right)z(x) + \frac{(k + 1)\,\varepsilon'}{\Delta_i^\ell(x)\prod_{k=\ell+1}^{L}\hat{\Delta}^k}\left(z^+(x) + z^-(x)\right) &\qquad \text{By Event (8) above} \\
= \left(1 + k\,\varepsilon_\ell^2\right)z(x) + \frac{(k + 1)\,\varepsilon'}{\prod_{k=\ell+1}^{L}\hat{\Delta}^k}\,|z(x)| &\qquad \text{By } \Delta_i^\ell(x) = \frac{z^+(x) + z^-(x)}{|z(x)|} \\
= \left(1 + k\,\varepsilon_\ell^2\right)z(x) + (k + 1)\,\varepsilon_{\ell+1}\,|z(x)|.
\end{aligned}
$$

To upper bound the last expression above, we begin by observing that $k\varepsilon_\ell^2 \leq \varepsilon_\ell$, which follows from the fact that $\varepsilon_\ell \leq \frac{1}{2\,(L-1)} \leq \frac{1}{k}$ by definition. Moreover, we also note that $\varepsilon_\ell \leq \varepsilon_{\ell+1}$ by definition of $\hat{\Delta}^\ell \geq 1$.

Now, we consider two cases.

**Case of $z(x) \geq 0$:** In this case, we have

$$
\begin{aligned}
\hat{z}(x) &\leq \left(1 + k\,\varepsilon_\ell^2\right)z(x) + (k + 1)\,\varepsilon_{\ell+1}\,|z(x)| \\
&\leq (1 + \varepsilon_\ell)z(x) + (k + 1)\varepsilon_{\ell+1}z(x) \\
&\leq (1 + \varepsilon_{\ell+1})z(x) + (k + 1)\varepsilon_{\ell+1}z(x) \\
&= (1 + (k + 2)\,\varepsilon_{\ell+1})\,z(x) \\
&= (1 + 2\,(\ell - 1)\varepsilon_{\ell+1})\,z(x),
\end{aligned}
$$

where the last line follows by definition of $k = 2\,(\ell - 2)$, which implies that $k + 2 = 2(\ell - 1)$. Thus, this establishes the desired upper bound in the case that $z(x) \geq 0$.

**Case of $z(x) < 0$:** Since $z(x)$ is negative, we have that $\left(1 + k\,\varepsilon_\ell^2\right)z(x) \leq z(x)$ and $|z(x)| = -z(x)$ and thus

$$
\begin{aligned}
\hat{z}(x) &\leq \left(1 + k\,\varepsilon_\ell^2\right)z(x) + (k + 1)\,\varepsilon_{\ell+1}\,|z(x)| \\
&\leq z(x) - (k + 1)\varepsilon_{\ell+1}z(x) \\
&\leq (1 - (k + 1)\varepsilon_{\ell+1})\,z(x) \\
&\leq (1 - (k + 2)\varepsilon_{\ell+1})\,z(x) \\
&= (1 - 2\,(\ell - 1)\varepsilon_{\ell+1})\,z(x),
\end{aligned}
$$

and this establishes the upper bound for the case of $z(x)$ being negative.

Putting the results of the case by case analysis together, we have the upper bound of $\hat{z}(x) \leq z(x) + 2\,(\ell - 1)\varepsilon_{\ell+1}|z(x)|$. The proof for establishing the lower bound for $z(x)$ is analogous to that given above, and yields $\hat{z}(x) \geq z(x) - 2\,(\ell - 1)\varepsilon_{\ell+1}|z(x)|$. Putting both the upper and lower bound together, we have that with probability at least $1 - \frac{\delta}{\eta}$:

$$\hat{z}(x) \in (1 \pm 2\,(\ell - 1)\varepsilon_{\ell+1})\,z(x),$$

and this completes the proof.

$\square$

### A.2.4 Remarks on Negative Activations

We note that up to now we assumed that the input $a(x)$, i.e., the activations from the previous layer, are strictly nonnegative. For layers $\ell \in \{3, \ldots, L\}$, this is indeed true due to the nonnegativity of the ReLU activation function. For layer 2, the input is $a(x) = x$, which can be decomposed into $a(x) = a_{\mathrm{pos}}(x) - a_{\mathrm{neg}}(x)$, where $a_{\mathrm{pos}}(x) \geq 0 \in \mathbb{R}^{\eta^{\ell-1}}$ and $a_{\mathrm{neg}}(x) \geq 0 \in \mathbb{R}^{\eta^{\ell-1}}$. Furthermore, we can define the sensitivity over the set of points $\{a_{\mathrm{pos}}(x), a_{\mathrm{neg}}(x) \mid x \in \mathcal{S}\}$ (instead of $\{a(x) \mid x \in \mathcal{S}\}$), and thus maintain the required nonnegativity of the sensitivities. Then, in the terminology of Lemma 2, we let

$$z_{\mathrm{pos}}^{+}(x) = \sum_{k \in \mathcal{W}_{+}} w_k\, a_{\mathrm{pos},k}(x) \geq 0 \qquad \text{and} \qquad z_{\mathrm{neg}}^{-}(x) = \sum_{k \in \mathcal{W}_{-}} (-w_k)\, a_{\mathrm{neg},k}(x) \geq 0$$

be the corresponding positive parts, and

$$z_{\mathrm{neg}}^{+}(x) = \sum_{k \in \mathcal{W}_{+}} w_k\, a_{\mathrm{neg},k}(x) \geq 0 \qquad \text{and} \qquad z_{\mathrm{pos}}^{-}(x) = \sum_{k \in \mathcal{W}_{-}} (-w_k)\, a_{\mathrm{pos},k}(x) \geq 0$$

be the corresponding negative parts of the preactivation of the considered layer, such that

$$z^{+}(x) = z_{\mathrm{pos}}^{+}(x) + z_{\mathrm{neg}}^{-}(x) \qquad \text{and} \qquad z^{-}(x) = z_{\mathrm{neg}}^{+}(x) + z_{\mathrm{pos}}^{-}(x).$$

We also let

$$\Delta_{i}^{\ell}(x) = \frac{z^{+}(x) + z^{-}(x)}{|z(x)|}$$

be as before, with $z^{+}(x)$ and $z^{-}(x)$ defined as above. Equipped with above definitions, we can rederive Lemma 2 analogously in the more general setting, i.e., with potentially negative activations. We also note that we require a slightly larger sample size now since we have to take a union bound over the failure probabilities of all four approximations (i.e. $\hat{z}_{\mathrm{pos}}^{+}(x)$, $\hat{z}_{\mathrm{neg}}^{-}(x)$, $\hat{z}_{\mathrm{neg}}^{+}(x)$, and $\hat{z}_{\mathrm{pos}}^{-}(x)$) to obtain the desired overall failure probability of $\delta/\eta$.

### A.2.5 Proof of Theorem 4

The following corollary immediately follows from Lemma 2 and establishes a layer-wise approximation guarantee.

**Corollary 9** (Conditional Layer-wise Approximation)**.** *Let $\varepsilon, \delta \in (0,1)$, $\ell \in \{2, \ldots, L\}$, and $x \sim \mathcal{D}$.* CORENET *generates a sparse weight matrix $\hat{W}^{\ell} = \left(\hat{w}_1^{\ell}, \ldots, \hat{w}_{\eta^{\ell}}^{\ell}\right)^{\top} \in \mathbb{R}^{\eta^{\ell} \times \eta^{\ell-1}}$ such that*

$$\mathbb{P}(\mathcal{E}^{\ell} \mid \mathcal{E}^{\ell-1}) = \mathbb{P}\left(\hat{z}^{\ell}(x) \in (1 \pm 2\,(\ell - 1)\,\varepsilon_{\ell+1})\,z^{\ell}(x) \mid \mathcal{E}^{\ell-1}\right) \geq 1 - \frac{\delta\,\eta^{\ell}}{\eta}, \tag{9}$$

*where $\varepsilon_{\ell} = \frac{\varepsilon'}{\widehat{\Delta}^{\ell \to}}$, $\hat{z}^{\ell}(x) = \hat{W}^{\ell}\hat{a}^{\ell}(x)$, and $z^{\ell}(x) = W^{\ell}a^{\ell}(x)$.*

*Proof.* Since (1) established by Lemma 2 holds for any neuron $i \in [\eta^{\ell}]$ in layer $\ell$ and since $(\mathcal{E}^{\ell})^{\mathsf{c}} = \cup_{i \in [\eta^{\ell}]}(\mathcal{E}_i^{\ell})^{\mathsf{c}}$, it follows by the union bound over the failure events $(\mathcal{E}_i^{\ell})^{\mathsf{c}}$ for all $i \in [\eta^{\ell}]$ that with probability at least $1 - \frac{\eta^{\ell}\delta}{\eta}$

$$\hat{z}^{\ell}(x) = \hat{W}^{\ell}\hat{a}^{\ell-1}(x) \in (1 \pm 2\,(\ell - 1)\,\varepsilon_{\ell+1})\,W^{\ell}a^{\ell-1}(x) = (1 \pm 2\,(\ell - 1)\,\varepsilon_{\ell+1})\,z^{\ell}(x).$$

$\square$

The following lemma removes the conditioning on $\mathcal{E}^{\ell-1}$ and explicitly considers the (compounding) error incurred by generating coresets $\hat{W}^2, \ldots, \hat{W}^\ell$ for multiple layers.

**Lemma 3** (Layer-wise Approximation). *Let $\varepsilon, \delta \in (0, 1)$, $\ell \in \{2, \ldots, L\}$, and $x \sim \mathcal{D}$. CORENET generates a sparse weight matrix $\hat{W}^\ell \in \mathbb{R}^{\eta^\ell \times \eta^{\ell-1}}$ such that, for $\hat{z}^\ell(x) = \hat{W}^\ell \hat{a}^\ell(x)$,*

$$\mathop{\mathbb{P}}_{(\hat{W}^2, \ldots, \hat{W}^\ell),\, x}(\mathcal{E}^\ell) = \mathop{\mathbb{P}}_{(\hat{W}^2, \ldots, \hat{W}^\ell),\, x}\left(\hat{z}^\ell(x) \in (1 \pm 2\,(\ell-1)\,\varepsilon_{\ell+1})\,z^\ell(x)\right) \geq 1 - \frac{\delta \sum_{\ell'=2}^\ell \eta^{\ell'}}{\eta}.$$

*Proof.* Invoking Corollary 9, we know that for any layer $\ell' \in \{2, \ldots, L\}$,

$$\mathop{\mathbb{P}}_{\hat{W}^{\ell'},\, x,\, \hat{a}^{\ell'-1}(\cdot)}(\mathcal{E}^{\ell'} \mid \mathcal{E}^{\ell'-1}) \geq 1 - \frac{\delta\,\eta^{\ell'}}{\eta}. \tag{10}$$

We also have by the law of total probability that

$$\begin{aligned}
\mathbb{P}(\mathcal{E}^{\ell'}) &= \mathbb{P}(\mathcal{E}^{\ell'} \mid \mathcal{E}^{\ell'-1})\,\mathbb{P}(\mathcal{E}^{\ell'-1}) + \mathbb{P}(\mathcal{E}^{\ell'} \mid (\mathcal{E}^{\ell'-1})^{\mathsf{c}})\,\mathbb{P}((\mathcal{E}^{\ell'-1})^{\mathsf{c}}) \\
&\geq \mathbb{P}(\mathcal{E}^{\ell'} \mid \mathcal{E}^{\ell'-1})\,\mathbb{P}(\mathcal{E}^{\ell'-1})
\end{aligned} \tag{11}$$

Repeated applications of (10) and (11) in conjunction with the observation that $\mathbb{P}(\mathcal{E}^1) = 1$[4] yield

$$\begin{aligned}
\mathbb{P}(\mathcal{E}^\ell) &\geq \mathbb{P}(\mathcal{E}^{\ell'} \mid \mathcal{E}^{\ell'-1})\,\mathbb{P}(\mathcal{E}^{\ell'-1}) \\
&\;\;\vdots && \text{Repeated applications of (11)} \\
&\geq \prod_{\ell'=2}^\ell \mathbb{P}(\mathcal{E}^{\ell'} \mid \mathcal{E}^{\ell'-1}) \\
&\geq \prod_{\ell'=2}^\ell \left(1 - \frac{\delta\,\eta^{\ell'}}{\eta}\right) && \text{By (10)} \\
&\geq 1 - \frac{\delta}{\eta} \sum_{\ell'=2}^\ell \eta^{\ell'} && \text{By the Weierstrass Product Inequality,}
\end{aligned}$$

where the last inequality follows by the Weierstrass Product Inequality[5] and this establishes the lemma. $\qquad\square$

Appropriately invoking Lemma 3, we can now establish the approximation guarantee for the entire neural network. This is stated in Theorem 4 and the proof can be found below.

**Theorem 4** (Network Compression). *For $\varepsilon, \delta \in (0, 1)$, Algorithm 1 generates a set of parameters $\hat{\theta} = (\hat{W}^2, \ldots, \hat{W}^L)$ of size*

$$\mathrm{nnz}(\hat{\theta}) \leq \sum_{\ell=2}^L \sum_{i=1}^{\eta^\ell} \left(\left\lceil \frac{32\,(L-1)^2\,(\hat{\Delta}^{\ell\to})^2\,S_i^\ell\,\log(\eta\,\eta^*)\,\log(8\,\eta/\delta)}{\varepsilon^2} \right\rceil + 1\right)$$

*in $\mathcal{O}\left(\eta\,\eta^*\,\log\left(\eta\,\eta^*/\delta\right)\right)$ time such that $\mathbb{P}_{\hat{\theta},\, x \sim \mathcal{D}}\left(f_{\hat{\theta}}(x) \in (1 \pm \varepsilon) f_\theta(x)\right) \geq 1 - \delta$.*

---

[4]Since we do not compress the input layer.

[5]The Weierstrass Product Inequality (Doerr, 2018) states that for $p_1, \ldots, p_n \in [0, 1]$,

$$\prod_{i=1}^n (1 - p_i) \geq 1 - \sum_{i=1}^n p_i.$$

*Proof.* Invoking Lemma 3 with $\ell = L$, we have that for $\hat{\theta} = (\hat{W}^2, \ldots, \hat{W}^L)$,

$$\mathop{\mathbb{P}}_{\hat{\theta}, x} \left( f_{\hat{\theta}}(x) \in 2 \left( L - 1 \right) \varepsilon_{L+1} f_{\theta}(x) \right) = \mathop{\mathbb{P}}_{\hat{\theta}, x} \left( \hat{z}^L(x) \in 2 \left( L - 1 \right) \varepsilon_{L+1} z^L(x) \right)$$

$$= \mathbb{P}(\mathcal{E}^L)$$

$$\geq 1 - \frac{\delta \sum_{\ell'=2}^{L} \eta^{\ell'}}{\eta}$$

$$= 1 - \delta,$$

where the last equality follows by definition of $\eta = \sum_{\ell=2}^{L} \eta^{\ell}$. Note that by definition,

$$\varepsilon_{L+1} = \frac{\varepsilon}{2 \left( L - 1 \right) \prod_{k=L+1}^{L} \hat{\Delta}^k}$$

$$= \frac{\varepsilon}{2 \left( L - 1 \right)},$$

where the last equality follows by the fact that the empty product $\prod_{k=L+1}^{L} \hat{\Delta}^k$ is equal to 1.

Thus, we have

$$2 \left( L - 1 \right) \varepsilon_{L+1} = \varepsilon,$$

and so we conclude

$$\mathop{\mathbb{P}}_{\hat{\theta}, x} \left( f_{\hat{\theta}}(x) \in (1 \pm \varepsilon) f_{\theta}(x) \right) \geq 1 - \delta,$$

which, along with the sampling complexity of Alg. 2 (Line 6), establishes the approximation guarantee provided by the theorem.

For the computational time complexity, we observe that the most time consuming operation per iteration of the loop on Lines 7-13 is the weight sparsification procedure. The asymptotic time complexity of each SPARSIFY invocation for each neuron $i \in [\eta^{\ell}]$ in layers $\ell \in \{2, \ldots, L\}$ (Alg. 1, Line 12) is dominated by the relative importance computation for incoming edges (Alg. 2, Lines 1-2). This can be done by evaluating $w_{ik}^{\ell} a_k^{\ell-1}(x)$ for all $k \in \mathcal{W}$ and $x \in \mathcal{S}$, for a total computation time that is bounded above by $\mathcal{O}\left( |\mathcal{S}| \, \eta^{\ell-1} \right)$ since $|\mathcal{W}| \leq \eta^{\ell-1}$ for each $i \in [\eta^{\ell}]$. Thus, SPARSIFY takes $\mathcal{O}\left( |\mathcal{S}| \, \eta^{\ell-1} \right)$ time. Summing the computation time over all layers and neurons in each layer, we obtain an asymptotic time complexity of $\mathcal{O}\left( |\mathcal{S}| \sum_{\ell=2}^{L} \eta^{\ell-1} \eta^{\ell} \right) \subseteq \mathcal{O}\left( |\mathcal{S}| \, \eta^* \, \eta \right)$. Since $|\mathcal{S}| \in \mathcal{O}(\log(\eta \, \eta^*/\delta))$, we conclude that the computational complexity our neural network compression algorithm is

$$\mathcal{O}\left( \eta \, \eta^* \, \log \left( \eta \, \eta^*/\delta \right) \right). \tag{12}$$

$\square$

### A.2.6 PROOF OF THEOREM 11

In order to ensure that the established sampling bounds are non-vacuous in terms of the sensitivity, i.e., not linear in the number of incoming edges, we show that the sum of sensitivities per neuron $S$ is small. The following lemma establishes that the sum of sensitivities can be bounded *instance-independent* by a term that is logarithmic in roughly the total number of edges $(\eta \cdot \eta^*)$.

**Lemma 10** (Sensitivity Bound). *For any $\ell \in \{2, \ldots, L\}$ and $i \in [\eta^{\ell}]$, the sum of sensitivities $S = S_+ + S_-$ is bounded by*

$$S \leq 2 \, |\mathcal{S}| = 2 \lceil \log \left( 8 \, \eta \, \eta^*/\delta \right) \log(\eta \, \eta^*) \rceil.$$

*Proof.* Consider $S_+$ for an arbitrary $\ell \in \{2, \ldots, L\}$ and $i \in [\eta^{\ell}]$. For all $j \in \mathcal{W}$ we have the following bound on the sensitivity of a single $j \in \mathcal{W}$,

$$s_j = \max_{x \in \mathcal{S}} g_j(x) \leq \sum_{x \in \mathcal{S}} g_j(x) = \sum_{x \in \mathcal{S}} \frac{w_j \, a_j(x)}{\sum_{k \in \mathcal{W}} w_k \, a_k(x)},$$

where the inequality follows from the fact that we can upper bound the max by a summation over $x \in \mathcal{S}$ since $g_j(x) \geq 0, \forall j \in \mathcal{W}$. Thus,

$$
\begin{aligned}
S_+ = \sum_{j \in \mathcal{W}} s_j &\leq \sum_{j \in \mathcal{W}} \sum_{x \in \mathcal{S}} g_j(x) \\
&= \sum_{x \in \mathcal{S}} \frac{\sum_{j \in \mathcal{W}} w_j \, a_j(x)}{\sum_{k \in \mathcal{W}} w_k \, a_k(x)} = |\mathcal{S}|,
\end{aligned}
$$

where we used the fact that the sum of sensitivities is finite to swap the order of summation.

Using the same argument as above, we obtain $S_- = \sum_{j \in \mathcal{W}_-} s_j \leq |\mathcal{S}|$, which establishes the lemma.
$\square$

Note that the sampling complexities established above have a linear dependence on the sum of sensitivities, $\sum_{\ell=2}^{L} \sum_{i=1}^{\eta^\ell} S_i^\ell$, which is instance-dependent, i.e., depends on the sampled $\mathcal{S} \subseteq \mathcal{P}$ and the actual weights of the trained neural network. By applying Lemma 10, we obtain a bound on the size of the compressed network that is independent of the sensitivity.

**Theorem 11** (Sensitivity-Independent Network Compression). *For any given $\varepsilon, \delta \in (0,1)$ our sampling scheme (Alg. 1) generates a set of parameters $\hat{\theta}$ of size*

$$
\mathrm{nnz}(\hat{\theta}) \in \mathcal{O}\left( \frac{\log(\eta/\delta) \, \log(\eta \, \eta^*/\delta) \log^2(\eta \, \eta^*) \, \eta \, L^2}{\varepsilon^2} \sum_{\ell=2}^{L} (\hat{\Delta}^{\ell \to})^2 \right),
$$

*in $\mathcal{O}\left( \eta \, \eta^* \log\left(\eta \, \eta^*/\delta\right) \right)$ time, such that $\mathbb{P}_{\hat{\theta}, \, x \sim \mathcal{D}}\left( f_{\hat{\theta}}(x) \in (1 \pm \varepsilon) f_\theta(x) \right) \geq 1 - \delta.$*

*Proof.* Combining Lemma 10 and Theorem 4 establishes the theorem.
$\square$

### A.2.7 GENERALIZED NETWORK COMPRESSION

Theorem 4 gives us an approximation guarantee with respect to one randomly drawn point $x \sim \mathcal{D}$. The following corollary extends this approximation guarantee to any set of $n$ randomly drawn points using a union bound argument, which enables approximation guarantees for, e.g., a test data set composed of $n$ i.i.d. points drawn from the distribution. We note that the sampling complexity only increases by roughly a logarithmic term in $n$.

**Corollary 12** (Generalized Network Compression). *For any $\varepsilon, \delta \in (0,1)$ and a set of i.i.d. input points $\mathcal{P}'$ of cardinality $|\mathcal{P}'| \in \mathbb{N}_+$, i.e., $\mathcal{P}' \overset{i.i.d.}{\sim} \mathcal{D}^{|\mathcal{P}'|}$, consider the reparameterized version of Alg. 1 with*

1. *$\mathcal{S} \subseteq \mathcal{P}$ of size $|\mathcal{S}| \geq \lceil \log\left(16 \, |\mathcal{P}'| \, \eta \, \eta^*/\delta\right) \log(\eta \, \eta^*) \rceil$,*

2. *$\hat{\Delta}^\ell = \left( \frac{1}{|\mathcal{S}|} \max_{i \in [\eta^\ell]} \sum_{x' \in \mathcal{S}} \Delta_i^\ell(x') \right) + \kappa$ as before, but $\kappa$ is instead defined as*

$$
\kappa = \sqrt{2\lambda_*} \left( 1 + \sqrt{2\lambda_*} \log\left(16 \, |\mathcal{P}'| \, \eta \, \eta^*/\delta\right) \right), \qquad \text{and}
$$

3. *$m \geq \left\lceil \frac{8 \, S \, \log(\eta \, \eta^*) \log(16 |\mathcal{P}'| \, \eta/\delta)}{\varepsilon_\ell^2} \right\rceil$ in the sample complexity in SPARSIFYWEIGHTS.*

*Then, Alg. 1 generates a set of neural network parameters $\hat{\theta}$ of size at most*

$$
\begin{aligned}
\mathrm{nnz}(\hat{\theta}) &\leq \sum_{\ell=2}^{L} \sum_{i=1}^{\eta^\ell} \left( \left\lceil \frac{32 \, (L-1)^2 \, (\hat{\Delta}^{\ell \to})^2 \, S_i^\ell \, \log(\eta \, \eta^*) \, \log(16 \, |\mathcal{P}'| \, \eta/\delta)}{\varepsilon^2} \right\rceil + 1 \right) \\
&\in \mathcal{O}\left( \frac{\log(\eta \, \eta^*) \, \log(\eta \, |\mathcal{P}'|/\delta) \, L^2}{\varepsilon^2} \sum_{\ell=2}^{L} (\hat{\Delta}^{\ell \to})^2 \sum_{i=1}^{\eta^\ell} S_i^\ell \right),
\end{aligned}
$$

*in $\mathcal{O}\left(\eta\,\eta^*\,\log\left(\eta\,\eta^*\,|\mathcal{P}'|/\delta\right)\right)$ time such that*

$$\mathbb{P}_{\hat{\theta},\,x}\left(\forall x \in \mathcal{P}' : f_{\hat{\theta}}(x) \in (1 \pm \varepsilon)f_{\theta}(x)\right) \geq 1 - \frac{\delta}{2}.$$

*Proof.* The reparameterization enables us to invoke Theorem 4 with $\delta' = {\delta}/{2}\,|\mathcal{P}'|$; applying the union bound over all $|\mathcal{P}'|$ i.i.d. samples in $\mathcal{P}'$ establishes the corollary. $\qquad\square$

## B  ADDITIONAL RESULTS

In this section, we give more details on the evaluation of our compression algorithm on popular benchmark data sets and varying fully-connected neural network configurations. In the experiments, we compare the effectiveness of our sampling scheme in reducing the number of non-zero parameters of a network to that of uniform sampling and the singular value decomposition (SVD). All algorithms were implemented in Python using the PyTorch library (Paszke et al., 2017) and simulations were conducted on a computer with a 2.60 GHz Intel i9-7980XE processor (18 cores total) and 128 GB RAM.

For training and evaluating the algorithms considered in this section, we used the following off-the-shelf data sets:

- *MNIST* (LeCun et al., 1998) — $70,000$ images of handwritten digits between 0 and 9 in the form of $28 \times 28$ pixels per image.

- *CIFAR-10* (Krizhevsky & Hinton, 2009) — $60,000$ $32 \times 32$ color images, a subset of the larger CIFAR-100 dataset, each depicting an object from one of 10 classes, e.g., airplanes.

- *FashionMNIST* (Xiao et al., 2017) — A recently proposed drop-in replacement for the MNIST data set that, like MNIST, contains $60,000$, $28 \times 28$ grayscale images, each associated with a label from 10 different categories.

We considered a diverse set of network configurations for each of the data sets. We varied the number of hidden layers between 2 and 5 and used either a constant width across all hidden layers between 200 and 1000 or a linearly decreasing width (denoted by "Pyramid" in the figures). Training was performed for 30 epochs on the normalized data sets using an Adam optimizer with a learning rate of 0.001 and a batch size of 300. The test accuracies were roughly 98% (MNIST), 45% (CIFAR10), and 96% (FashionMNIST), depending on the network architecture. To account for the randomness in the training procedure, for each data set and neural network configuration, we averaged our results across 4 trained neural networks.

### B.1  DETAILS ON THE COMPRESSION ALGORITHMS

We evaluated and compared the performance of the following algorithms on the aforementioned data sets.

1. *Uniform (Edge) Sampling* — A uniform distribution is used, rather than our sensitivity-based importance sampling distribution, to sample the incoming edges to each neuron in the network. Note that like our sampling scheme, uniform sampling edges generates an unbiased estimator of the neuron value. However, unlike our approach which explicitly seeks to minimize estimator variance using the bounds provided by empirical sensitivity, uniform sampling is prone to exhibiting large estimator variance.

2. *Singular Value Decomposition* (SVD) — The (truncated) SVD decomposition is used to generate a low-rank (rank-$r$) approximation for each of the weight matrices $(\hat{W}^2, \dots, \hat{W}^L)$ to obtain the corresponding parameters $\hat{\theta} = (\hat{W}_r^2, \dots, \hat{W}_r^L)$ for various values of $r \in \mathbb{N}_+$. Unlike the compared sampling-based methods, SVD does not sparsify the weight matrices. Thus, to achieve fair comparisons of compression rates, we compute the size of the rank-$r$ matrices constituting $\hat{\theta}$ as,

$$\text{nnz}(\hat{\theta}) = \sum_{\ell=2}^{L} \sum_{i=1}^{r} \left( \text{nnz}(u_i^\ell) + \text{nnz}(v_i^\ell) \right),$$

where $W^\ell = U^\ell \Sigma^\ell (V^\ell)^\top$ for each $\ell \in \{2, \dots, L\}$, with $\sigma_1 \geq \sigma_2 \dots \geq \sigma_{\eta^{\ell-1}}$ and $u_i^\ell$ and $v_i^\ell$ denote the $i$th columns of $U^\ell$ and $V^\ell$ respectively.

3. $\ell_1$ *Sampling (Achlioptas et al., 2013)* — An entry-wise sampling distribution based on the ratio between the absolute value of a single entry and the (entry-wise) $\ell_1$ - norm of the weight matrix is computed, and the weight matrix is subsequently sparsified by sampling accordingly. In particular, entry $w_{ij}$ of some weight matrix $W$ is sampled with probability

$$p_{ij} = \frac{|w_{ij}|}{\|W\|_{\ell_1}},$$

and reweighted to ensure the unbiasedness of the resulting estimator.

4. $\ell_2$ *Sampling (Drineas & Zouzias, 2011)* — The entries $(i, j)$ of each weight matrix $W$ are sampled with distribution

$$p_{ij} = \frac{w_{ij}^2}{\|W\|_F^2},$$

where $\|\cdot\|_F$ is the Frobenius norm of $W$, and reweighted accordingly.

5. $\frac{\ell_1 + \ell_2}{2}$ *Sampling (Kundu & Drineas, 2014)* – The entries $(i, j)$ of each weight matrix $W$ are sampled with distribution

$$p_{ij} = \frac{1}{2} \left( \frac{w_{ij}^2}{\|W\|_F^2} + \frac{|w_{ij}|}{\|W\|_{\ell_1}} \right),$$

where $\|\cdot\|_F$ is the Frobenius norm of $W$, and reweighted accordingly. We note that Kundu & Drineas (2014) constitutes the current state-of-the-art in data-oblivious matrix sparsification algorithms.

6. *CoreNet* (Edge Sampling) — Our core algorithm for edge sampling shown in Alg. 2, but without the neuron pruning procedure.

7. *CoreNet+* (CoreNet & Neuron Pruning) — Our algorithm shown in Alg. 1 that includes the neuron pruning step.

8. *CoreNet++* (CoreNet+ & Amplification) — In addition to the features of *Corenet+*, multiple coresets $\mathcal{C}_1, \ldots, \mathcal{C}_\tau$ are constructed over $\tau \in \mathbb{N}_+$ trials, and the best one is picked by evaluating the empirical error on a subset $\mathcal{T} \subseteq \mathcal{P} \setminus \mathcal{S}$ (see Sec. 4 for details).

## B.2 Preserving the Output of a Neural Network

We evaluated the accuracy of our approximation by comparing the output of the compressed network with that of the original one and compute the $\ell_1$-norm of the relative error vector. We computed the error metric for both the uniform sampling scheme as well as our compression algorithm (Alg. 1). Our results were averaged over 50 trials, where for each trial, the relative approximation error was averaged over the entire test set. In particular, for a test set $\mathcal{P}_{\text{test}} \subseteq \mathbb{R}^d$ consisting of $d$ dimensional points, the average relative error of with respect to the $f_{\hat\theta}$ generated by each compression algorithm was computed as

$$\text{error}_{\mathcal{P}_{\text{test}}}(f_{\hat\theta}) = \frac{1}{|\mathcal{P}_{\text{test}}|} \sum_{x \in \mathcal{P}_{\text{test}}} \left\| f_{\hat\theta}(x) - f_\theta(x) \right\|_1.$$

Figures 3, 4, and 5 depict the average performance of the compared algorithms for various network architectures trained on MNIST, CIFAR-10, and FashionMNIST, respectively. Our algorithm is able to compress networks trained on MNIST and FashionMNIST to about 10% of their original size without significant loss of accuracy. On CIFAR-10, a compression rate of 50% yields classification results comparable to that of uncompressed networks. The shaded region corresponding to each curve represents the values within one standard deviation of the mean.

## B.3 Preserving the Classification Performance

We also evaluated the accuracy of our approximation by computing the loss of prediction accuracy on a test data set, $\mathcal{P}_{\text{test}}$. In particular, let $\text{acc}_{\mathcal{P}_{\text{test}}}(f_\theta)$ be the average accuracy of the neural network $f_\theta$, i.e,.

$$\text{acc}_{\mathcal{P}_{\text{test}}}(f_\theta) = \frac{1}{|\mathcal{P}_{\text{test}}|} \sum_{x \in \mathcal{P}_{\text{test}}} \mathbb{1} \left( \underset{i \in [\eta^L]}{\arg\max} f_\theta(x) \neq y(x) \right),$$

where $y(x)$ denotes the (true) label associated with $x$. Then the drop in accuracy is computed as

$$\text{acc}_{\mathcal{P}_{\text{test}}}(f_\theta) - \text{acc}_{\mathcal{P}_{\text{test}}}(f_{\hat\theta}).$$

Figures 6, 7, and 8 depict the average performance of the compared algorithms for various network architectures trained on MNIST, CIFAR-10, and FashionMNIST respectively. The shaded region corresponding to each curve represents the values within one standard deviation of the mean.

### B.4 PRELIMINARY RESULTS WITH RETRAINING

We compared the performance of our approach with that of the popular weight thresholding heuristic – henceforth denoted by WT – of Han et al. (2015) when retraining was allowed after the compression, i.e., pruning, procedure. Our comparisons with retraining for the networks and data sets mentioned in Sec. 6 are as follows. For MNIST, WT required 5.8% of the number of parameters to obtain the classification accuracy of the original model (i.e., 0% drop in accuracy), whereas for the same percentage (5.8%) of the parameters retained, CoreNet++ incurred a classification accuracy drop of 1%. For CIFAR, the approach of Han et al. (2015) matched the original models accuracy using 3% of the parameters, whereas CoreNet++ reported an accuracy drop of 9.5% for 3% of the parameters retained. Finally, for FashionMNIST, the corresponding numbers were 4.1% of the parameters to achieve 0% loss for WT, and a loss of 4.7% in accuracy for CoreNet++ with the same percentage of parameters retained.

### B.5 DISCUSSION

As indicated in Sec. 6, the simulation results presented here validate our theoretical results and suggest that empirical sensitivity can lead to effective, more informed sampling compared to other methods. Moreover, we are able to outperform networks that are compressed via state-of-the-art matrix sparsification algorithms. We also note that there is a notable difference in the performance of our algorithm between different datasets. In particular, the difference in performance of our algorithm compared to the other method for networks trained on FashionMNIST and MNIST is much more significant than for networks trained on CIFAR. We conjecture that this is partially due to considering only fully-connected networks as these network perform fairly poorly on CIFAR (around 45% classification accuracy) and thus edges have more uniformly distributed sensitivity as the information content in the network is limited. We envision that extending our guarantees to convolutional neural networks may enable us to further reason about the performance on data sets such as CIFAR.

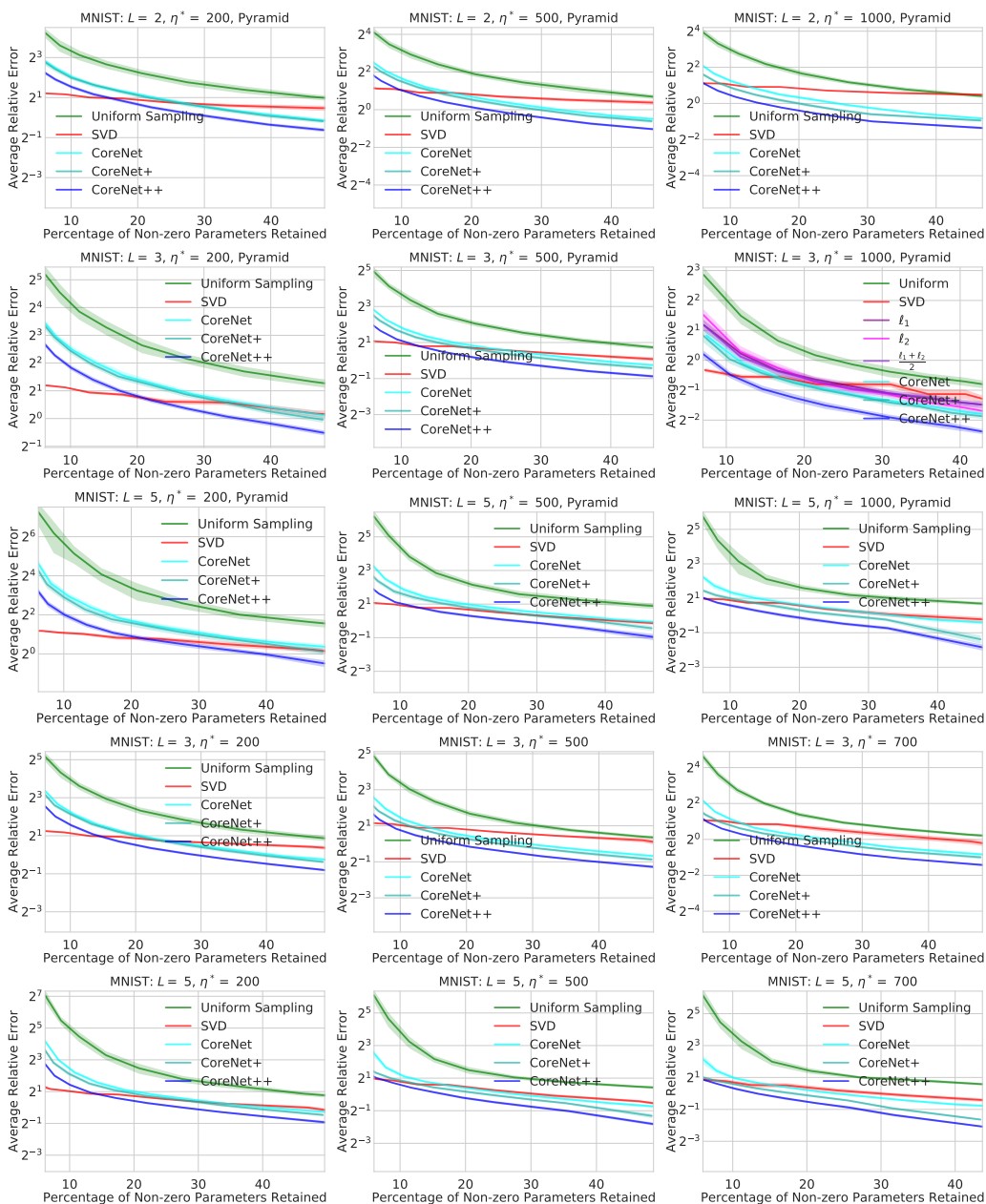

Figure 3: Evaluations against the MNIST dataset with varying number of hidden layers ($L$) and number of neurons per hidden layer ($\eta^*$). Shaded region corresponds to values within one standard deviation of the mean. The figures show that our algorithm's relative performance increases as the number of layers (and hence the number of redundant parameters) increases.

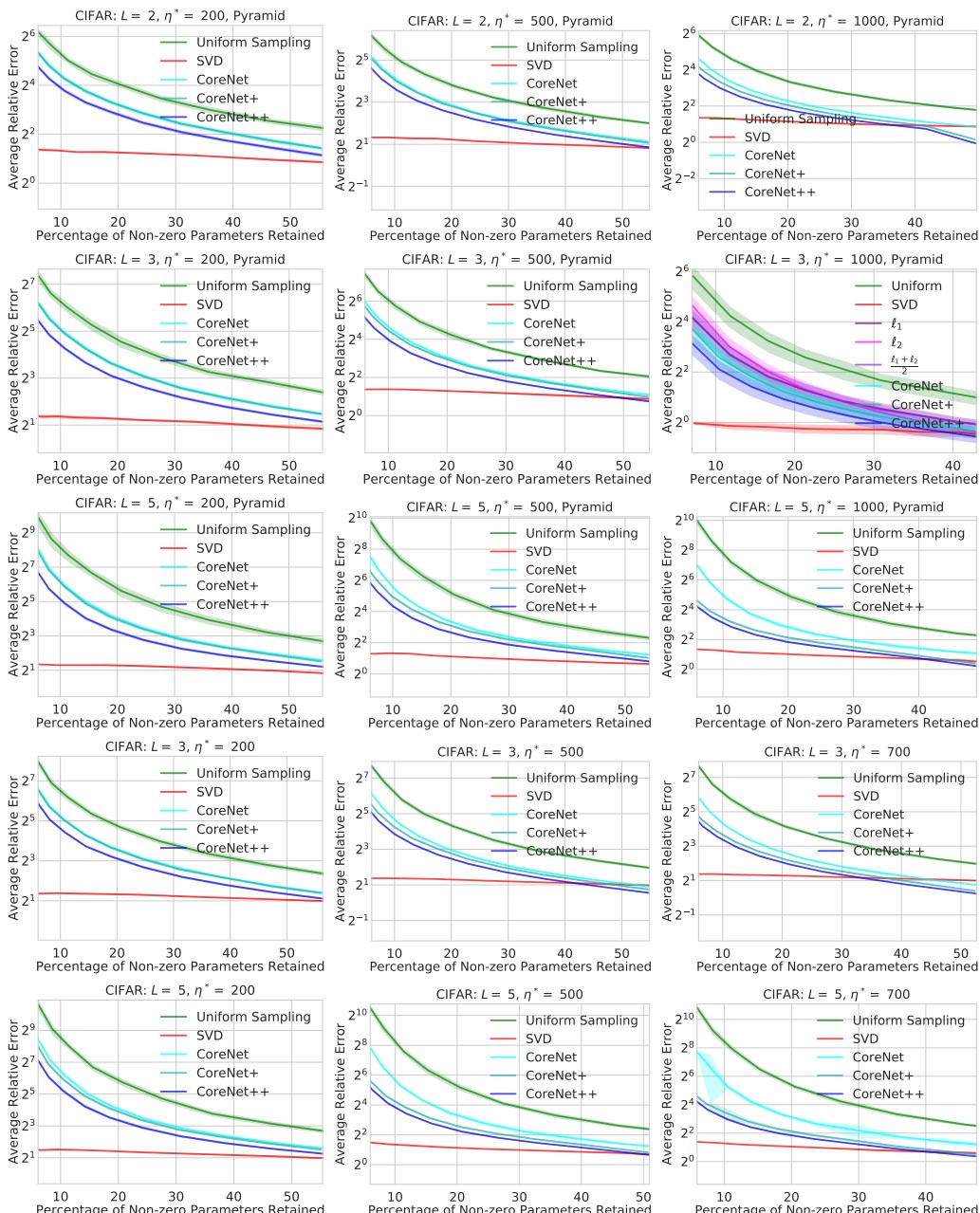

Figure 4: Evaluations against the CIFAR-10 dataset with varying number of hidden layers ($L$) and number of neurons per hidden layer ($\eta^*$). The trend of our algorithm's improved relative performance as the number of parameters increases (previously depicted in Fig. 3) also holds for the CIFAR-10 data set.

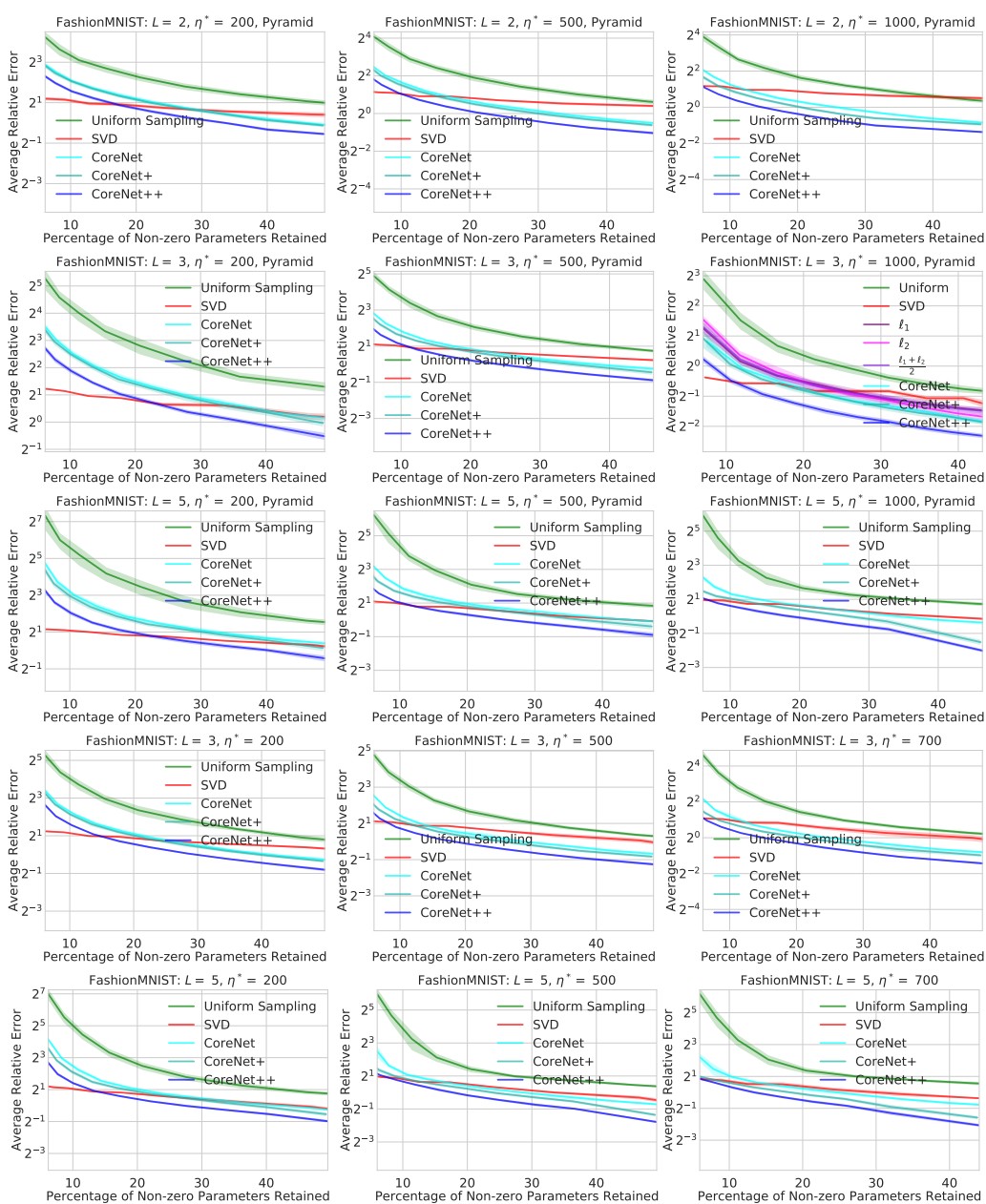

Figure 5: Evaluations against the FashionMNIST dataset with varying number of hidden layers ($L$) and number of neurons per hidden layer ($\eta^*$).

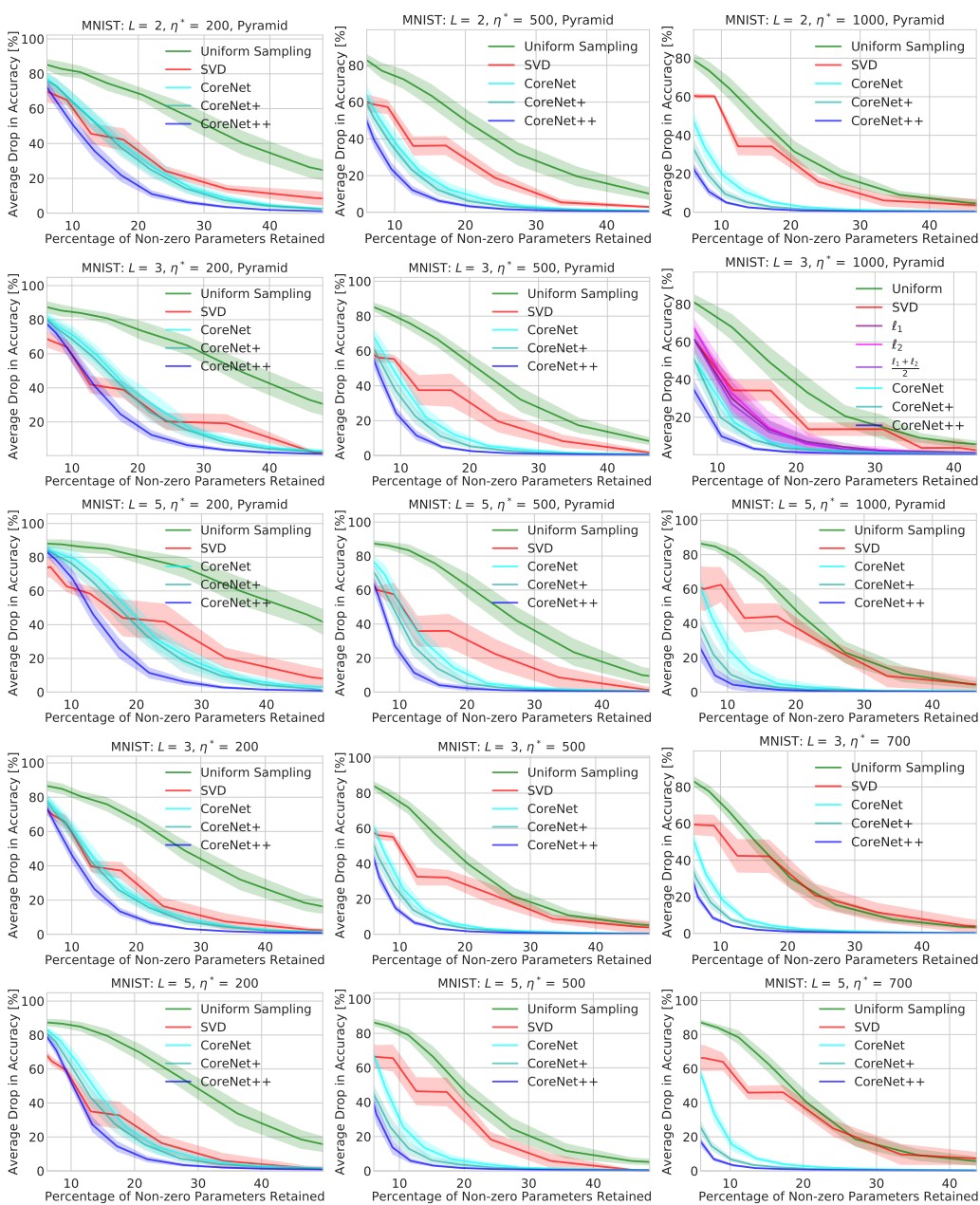

Figure 6: Evaluations against the MNIST dataset with varying number of hidden layers ($L$) and number of neurons per hidden layer ($\eta^*$). Shaded region corresponds to values within one standard deviation of the mean. The figures show that our algorithm's relative performance increases as the number of layers (and hence the number of redundant parameters) increases.

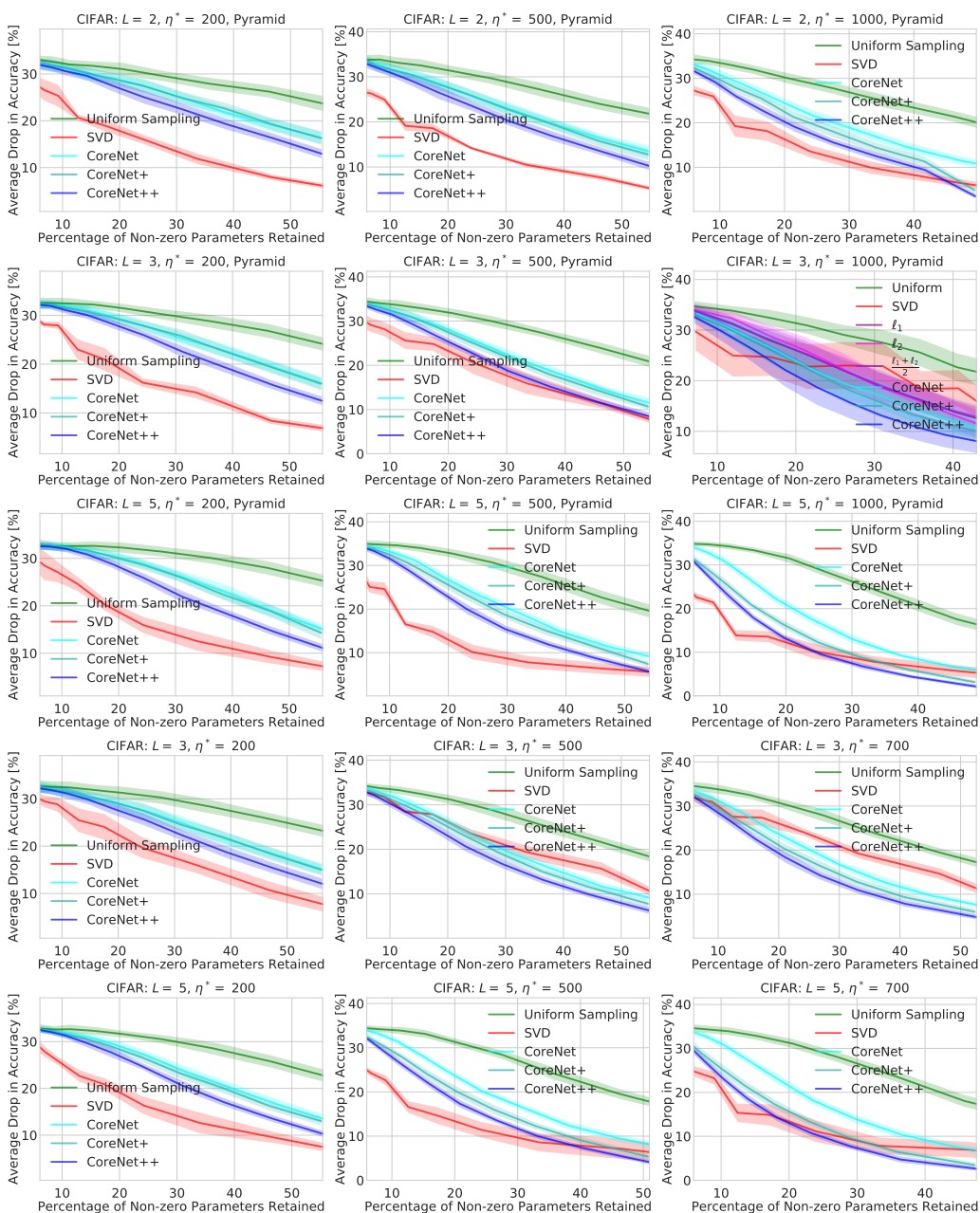

Figure 7: Evaluations against the CIFAR-10 dataset with varying number of hidden layers ($L$) and number of neurons per hidden layer ($\eta^*$). The trend of our algorithm's improved relative performance as the number of parameters increases (previously depicted in Fig. 6) also holds for the CIFAR-10 data set.

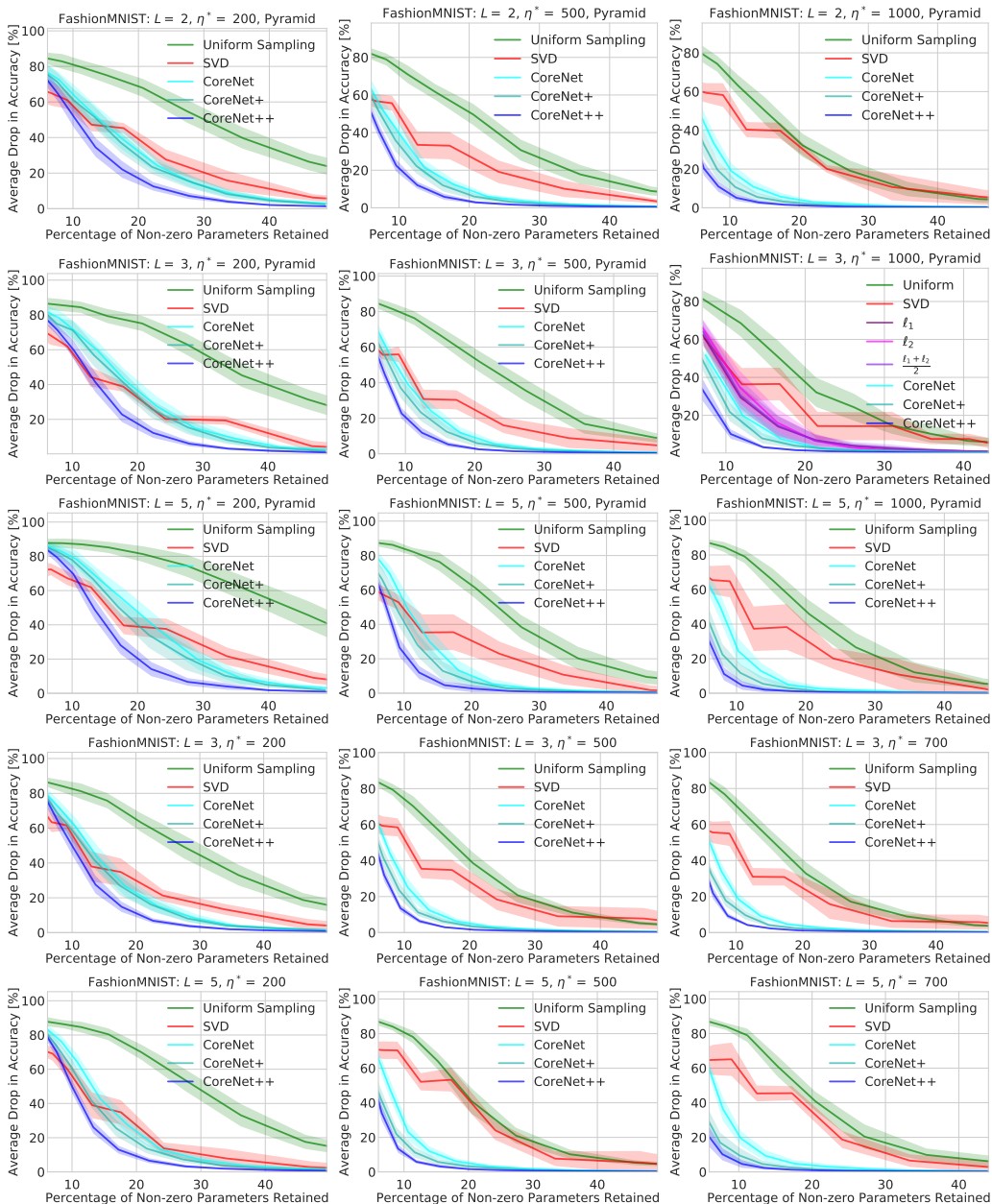

Figure 8: Evaluations against the FashionMNIST dataset with varying number of hidden layers ($L$) and number of neurons per hidden layer ($\eta^*$).

