# OpenReview forum: "Data-Dependent Coresets for Compressing Neural Networks with Applications to Generalization Bounds"
_ICLR.cc/2019/Conference_

### Official Review · AnonReviewer3 · 2018-10-29
**A subset of the input edges for each neuron is subsampled with probability proportional to the relative importance of each edge.**

**Rating:** 6
**Confidence:** 4

**Review:**

Given an additively decomposable function F(X, Q) = sum_over_x_in_X cost(x, Q), one can approximate it using either random sampling of x in X (unbiased, possibly high variance), or using importance sampling and replace the sum_over_x with a sum_over_coreset importance_of_a_point * cost(x, Q) which if properly defined can be both unbiased and have low variance [1]. In this work the authors consider the weighted sum of activations as F and suggest that for each neuron we can subsample the incoming edges. To construct the importance sampling strategy the authors adapt the classic notion of sensitivity from the coreset literature. Then, one has to carefully balance the approximation quality from one layer to the next and essentially union bound the results over all layers and all sampled points. The performed analysis is sound (up to my knowledge).

Pro:
- I commend the authors for a clean and polished writeup.
- The analysis seems to be sound (apart from the issues discussed below)
- The experimental results look promising, at least in the limited setup.

Con:
- There exists competing work with rigorous guarantees, for example [2].
- The analysis hinges on two assumptions which, in my opinion, make the problem feasible: having (sub) exponential tails allows for strong concentration results, and with proper analysis (as done by the authors), the fact that the additively decomposable function can be approximated given well-behaving summands is not surprising. The analysis is definitely non-trivial and I commend the authors for a clean writeup.
- While rigorous guarantees are lacking for some previous work, previously introduced techniques were shown to be extremely effective in practice and across a spectrum of tasks. As the guarantees arguably stem from the assumptions 1 and 2, I feel that it’s unfair to not compare to those results empirically. Hence, failing to compare to results of at least [2, 3] is a major drawback of this work.
- The result holds for n points drawn from P. However, in practice the network might receive essentially arbitrary input from P at inference time. Given that we need to decide on the number of edges to preserve apriori, what are the implications?
- The presented bounds should be discussed on an intuitive level (i.e. the number of non zero entries is approximately cubic in L).

I consider this to be a well-executed paper which brings together the main ideas from the coreset literature and shows one avenue of establishing provable results. However, given that no comparison to the state-of-the-art techniques is given I'm not confident that the community will apply these techniques in practice. On the other hand, the main strength -- the theoretical guarantees -- hinge on the introduced assumptions. As such, without additional empirical results demonstrating the utility with respect to the state-of-the-art methods (for the same capacity in terms of NNZ) I cannot recommend acceptance.

[1] https://arxiv.org/abs/1601.00617
[2] papers.nips.cc/paper/6910-net-trim-convex-pruning-of-deep-neural-networks-with-performance-guarantee
[3] https://arxiv.org/abs/1510.00149


========
Thank you for the detailed responses. Given the additional experimental results and connections to existing work, I have updated my score from 5 to 6.

---

> ### Author Response · Authors · 2018-11-16
> **Response to AnonReviewer3**
>
> We are grateful for the detailed and thorough review of our paper, and thank the reviewer for the constructive feedback.
>
> 1) Thank you for pointing out the related work [7]. We would like to highlight that [7] solves a convex optimization (in fact, a proxy to ||W||_0 by instead minimizing ||W||_1) to promote sparsity and consequently, in comparison to our work, does not give an explicit tradeoff between the resulting sparsity of the network and the approximation accuracy. Furthermore, the results of [7] only apply to approximating the output of the neural network with respect to the input training points X, whereas our compressed network provably approximates the output of the neural network for any point randomly drawn from the data distribution. Moreover, the bounds provided by the paper are for the link-normalized network where the matrices are normalized to have ell_1 norm equal to 1. This implies that, as noted by the authors, their error guarantees should be multiplied by ||W||_1 (which can be arbitrarily large) in order to map them back to appropriate guarantees for the original network.
>
> 2) Competing work, such as those mentioned by AnonReviewer1, also imposes assumptions (e.g., [7] as well as [9]) to ensure sufficiently small sampling complexity. We would like to emphasize that we impose Assumptions 1 and 2 solely to rule out pathological instances in which we cannot approximate the sensitivity of each edge or the \Delta of each neuron using a small-sized (~logarithmic in 1/\delta) set of data points S \subseteq P. If the desired failure probability \delta is sufficiently large and many data points are available for use in constructing S, then Assumptions 1 and 2 are not necessary. Furthermore, we believe it is important to highlight that our assumptions are satisfied for a variety of real-world data sets and quantities (e.g., for Asm. 2: all bounded random variables are subgaussian and hence subexponential) and distributions (e.g., for Asm. 1: traditional distributions such as uniform, normal, gamma, defined on [0,1] or [0, M] for M <= 1, among others, satisfy this assumption).
>
> We would also like to note that our assumptions can be made significantly milder and more general. In particular, the constant log(\eta \eta^*) for the upper bound of K in Assumption 1 and \lambda in Assumption 2 can be replaced by a general constant C > 1, and our sampling complexities (for the size of S and edge sampling complexity m in Alg. 1 and Alg. 2, respectively) would then simply be an expression containing C instead of log(\eta \eta^*).
>
> 3) Thank you for your reference to the related work. We would like to remark that [8] is based predominantly on heuristics and point out that the methods mentioned in the related work (such as [8]) are synergistic to our methods and can be used as a post- and/or preprocessing step in conjunction with our method. Furthermore, we would like to mention that the work of [8] is more concerned with reducing the storage requirements of the resulting compressed network (e.g., by Huffman coding), whereas our approach not only reduces storage requirements (by promoting sparsity), but also improves inference time complexity (via sparse linear algebra algorithms at inference time). We would like to investigate these prospective research directions and improvements to our algorithm in future work.
>
> 4) Finally, we would like to clarify that the number of points does not have to be fixed a priori. More generally speaking, our bound provides a probabilistic guarantee that any point that is input into the network is correctly approximated with probability 1-\delta. This holds for any randomly drawn data point. If, for example, we want to obtain an approximation guarantee for any set of n randomly drawn points, then taking \delta’ = \delta/n in our sampling complexity bounds in conjunction with a straightforward application of the union bound yields the desired approximation guarantee with probability at least 1 - \delta for the set of n points (see Corollary 12 - Generalized Network Compression, in the Appendix).
>
> [7]: papers.nips.cc/paper/6910-net-trim-convex-pruning-of-deep-neural-networks-with-performance-guarantee
> [8]: https://arxiv.org/abs/1510.00149
> [9]: "Near-optimal entrywise sampling for data matrices" by Achlioptas et al.

---

### Official Review · AnonReviewer1 · 2018-11-02
**principled approach to sparsification of neural network weights**

**Rating:** 7
**Confidence:** 4

**Review:**

The authors propose to reduce the size of fully connected neural networks, defined as the total number of nonzeros in the weight matrices, by calculating sensitivity scores for each incoming connection to a neuron, and randomly keeping only some of the incoming connections with probability proportional to their share of the total sensitivity. They provide a specific definition for the sensitivity scores and establish that the sparsified neural network, with constant probability for any sample from the training population, provides an output that is a small multiplicative factor away from the output of the unsparisfied neural network. The cost of the sparsification is essentially the application of the trained neural network to a small number of data points in order to compute the sensitivity scores

Pros:
- the method works empirically, in that their empirical evaluations on MNIST, CIFAR, and FashionMNIST classification problems show that the drop in accuracy is lower when the neural net is sparsified using their CoreNet algorithm and variations than when it is randomly sparsified or the neural network size is reduced by using SVD.
- theory is provided to argue the consistency of the sparsified neural network

Cons:
- no comparison is made to the baseline of using matrix sparsification algorithms on the weight matrices themselves. I do not see why CoreNet should be expected to perform empirically better than simply using e.g. the entry-wise sampling scheme from "Near-optimal entrywise sampling for data matrices" by Achlioptas and co-authors, or earlier works addressing the same problem of sparsifying matrices.
- the theory makes very strong assumptions (Assumptions 1 and 2) that are not explained or justified well. Both depend on the specific weight matrices being sparsified, and it isn't clear a priori when the weight matrices obtained from whatever optimization procedure was used to train the neural net will be such that these assumptions hold.
- despite the suggestions of the theory, the accuracy drop can be quite large in practice, as in the CIFAR panel of Figure 1

I think the ICLR audience will appreciate the attempt to provide a principled approach to decreasing the size of neural networks, but I do not think this approach is widely compelling as :
(1) no true guaranteed control on the trade-off between accuracy loss and network size is available
(2) empirically the method does not perform well consistently
(3) comparisons with reasonable and informative baselines are missing

Updated in response to author response: the inclusion of experimental comparisons with linear algebraic sparsification baselines, showing that the proposed method can be significantly more accurate, strengthens the appeal of the method.

---

> ### Author Response · Authors · 2018-11-16
> **Response to AnonReviewer1**
>
> We thank the reviewer for the in-depth review of our paper and the helpful reference to prior work on matrix sparsification by entrywise sampling.
>
> 1) The work of [5] on matrix sparsification is similar to our work in the sense that the aim is to approximate a weight matrix W by its sparse counterpart \hat W, but differs significantly from our end goal of generating \hat W to provably approximate the neuron’s value. In other words, the overarching goal of our compression is to approximate z = W*a entry-wise using a sparse matrix \hat W, i.e., \hat z = \hat W * a, whereas the focus of [6] is to compute a sparse \hat W such that the normed difference ||\hat W - W|| < epsilon. Without too much effort, one can see how, depending on the structure/distribution of the activation a, the weight-based sampling methods may fail for approximating the value z = W*a, despite the normed difference ||\hat W - W|| being small. Our insight is that rather than taking a data-oblivious approach to sparsification (i.e., simply generating \hat W that is close to W in some sense), we explicitly consider the input data distribution in our sensitivity computations to generate a more informed sampling distribution of the weight entries that specifically considers the end goal of approximating the neuron’s value. In other words, since we tailor the sparsification of the weight matrix to the underlying data distribution (using our new notion of empirical sensitivity), we obtain a more informed sparsification procedure that yields better performance. We refer the reader to our new figures in our revision that compare our algorithm to state-of-the-art entrywise matrix methods.
>
> We would also like to remark that [5] imposes a set of 3 “Data matrix” assumptions (see Definition 4.1 of [5]), which clearly do not hold for weight matrices of a neural network and thus render the sparsification approach of [5] inapplicable. Despite this, as mentioned above, we included comparisons to other state-of-the-art matrix sparsification methods that are based on entrywise sampling in our current revision.
>
> 2) We would like to highlight that these assumption rule out pathological instances in which we cannot approximate the sensitivity of each edge or the \Delta of each neuron using a small-sized set of data points S \subseteq P. In particular, we impose Assumption 1 to ensure that a subset of data points S of size roughly logarithmic in 1/delta can be used to obtain accurate approximations of the sensitivity of each edge. Assumption 2 is imposed to ensure that the same small sized set S can be used to approximate the quantity Delta, which represents the ratio of the positive and negative decompositions of the objective value z = <w,x>. Finally, we remark that defining the sampling complexity in terms of the Delta term is in line with related work, such as that of constructing coresets for logistic regression [6], where a complexity measure \mu(X) (analogous to our \Delta) is defined to be the maximum ratio of positive to negative contributions to the objective value and is used to quantify the sampling complexity. We will clarify the exposition of our assumptions and intuition behind our assumptions and better explain the intuition behind them in our final submission.
>
> 3) You are correct that the performance of the compression method varies depending on the data set. Nevertheless, competing compression methods (including the recently added comparisons) exhibit very similar performance variations between architectures and data sets. We believe this highlights the importance of the relation between the dataset and the network architecture, which is accurately reflected in our data-dependent and network-dependent sampling bounds, such as the sum of sensitivities and the value of Delta. We also believe that the fact that the sampling complexity can be determined on the fly by simply inspecting the pre-computed values of sensitivity (and considering their sum) and Delta is a strength of our approach. Since, these quantities can shed light on which parts/layers of a neural network are important to retain to preserve the network’s output, and in a sense, provide interpretability of the neural network’s components.
>
> 4) The trade-off between accuracy loss and network size can be readily computed from the available sampling complexity bounds of our entry-wise approximation guarantee (Lemma 3 & Theorem 4). In particular, the accuracy of the network is captured by the margin between the most likely output neuron and the remaining neurons. We can therefore pick a margin and obtain a bound in terms of the sampling complexity.
>
> 5) We would like to clarify that the failure probability of our algorithm is not constant, and in fact is exponentially small in the edge sample size and the size of the input points used to compute the sensitivity (S).
>
> [5]: "Near-optimal entrywise sampling for data matrices" by Achlioptas et al.
> [6]: https://arxiv.org/abs/1805.08571

---

### Official Review · AnonReviewer2 · 2018-11-03
**nice contribution -- guarantees+practice**

**Rating:** 6
**Confidence:** 3

**Review:**

In this work the authors improve upon the work of Arora et al. mainly with respect to one aspect, i.e.,
They provide eps-approximation of a fully connected neural network output neuron-wise. The idea of
compression is very natural and has been explored by various previous works (key refs are cited). Intuitively,
the number of effective parameters is significantly less than the number of parameters in the neural network.
The authors introduce the notion of the coreset that is suitable for compressing the weight parameters
in definition 1. Their main result is stated as Theorem 4. Finally, the authors experiment on standard benchmarks,
perform a careful experimental analysis (i.e., they ensure fairness of comparison between methods such as
SVD and the rest).  It would be interesting to see the histogram/distribution of the weights per layer and at an aggregate level
for the datasets used.  Also, in the light of the recent results of Arora et al. that show that the signal out of a layer
is correlated with the top singular values, how would coresets
developed in the numerical linear algebraic community  (e.g., Near-optimal Coresets For Least-Squares Regression
by Boutsidis et al.) perform, even as an experimental heuristic compared to the proposed method?

---

> ### Author Response · Authors · 2018-11-16
> **Response to AnonReviewer2**
>
> Thank you for your insightful comments and constructive feedback. Please find our specific comments below.
>
> 1) We would like to clarify that the focus of the paper is to provide a sampling-based compression technique using coresets that is simultaneously practical and provably correct. Towards this end, our coresets-based method provides entry-wise guarantees on the output of the neural network for points drawn from the input data distribution. As we noted in the related works section, this guarantee is significantly stronger than those of Arora et al. because their guarantees are only norm-based and only hold for the initial set of training points. This implies that their compressed network is not ensured to provably approximate the original net for, e.g., points coming from a test set, which significantly limits the applicability of their approach. Another difference to note is that Arora et al. use a JL-based approach whereas we use coresets to conduct the compression.
>
> Moreover, as we mentioned in our response to AnonReviewer1, our entry-wise guarantee enables the user to explicitly control the trade-off between classification accuracy loss and the network size, which is a functionality that norm-based bounds on the output cannot provide.
>
> 2) We agree that this is insightful to better understand the relationship between different network architectures, the data, and the compressibility of the network, and that including plots of these would be illuminating. In order to keep the length of our current submission in line with that of a conference paper, we intend to include the pertinent plots and a discussion about the distributions of the weights and the sensitivity of the parameters of each layer in future work.
>
> 3) Our revision contains comparisons to multiple state-of-the-art matrix sparsification techniques, as mentioned in our General Response. The focus of [4] is to reduce the amount of data required (data compression) to compute an approximately optimal least-squares solution, which significantly differs -- due to inherent differences in problem structure and objective functions -- from the problem of sparsifying the parameters of neural network matrices to approximately preserve the network’s output.
>
> [4]: Near-optimal Coresets For Least-Squares Regression by Boutsidis et al.

---

### Author Response · Authors · 2018-11-16
**General Response**

We thank all the reviewers for their useful suggestions and careful consideration of our paper. We believe that the quality and exposition of the paper has been significantly improved thanks to the reviewers’ feedback. Your feedback has raised several points regarding our contributions that we would like to clarify.

Our paper proposes a principled approach for provably compressing neural networks using an importance sampling scheme that is defined by leveraging a novel concept of empirical sensitivity. Constructing an importance sampling distribution using empirical sensitivity is particularly appealing because it is fast to compute using a small subset of the available data points and most importantly because it captures the relative importance of the weights w on the pre-activation value z = <w, a>. This enables us to analyze and bound the approximation guarantee of our sparsification algorithm with respect to the desired output z, rather than providing norm-based matrix bounds -- as prior approaches such as those based on SVD and/or other matrix sparsification methods do [1,2,3].

We are also thankful for the references to additional baseline and state-of-the-art methods provided by the reviewers. Our revision contains the results of additional experiments that compare the performance of our coreset-based algorithm to three state-of-the-art matrix sparsification algorithms with provable guarantees [1,2,3].

[1]: Drineas, Petros, and Anastasios Zouzias. "A note on element-wise matrix sparsification via a matrix-valued Bernstein inequality." Information Processing Letters 111.8 (2011): 385-389.
[2]: Achlioptas, Dimitris, Zohar Karnin, and Edo Liberty. "Matrix entry-wise sampling: Simple is best." KDD 2013.1.1 (2013): 1-4.
[3]: Kundu, Abhisek, and Petros Drineas. "A note on randomized element-wise matrix sparsification." arXiv preprint arXiv:1404.0320 (2014). https://arxiv.org/abs/1404.0320

---

### Meta-Review · Area_Chair1 · 2018-12-17

**Confidence:** 5
**Recommendation:** Accept (Poster)

**Metareview:**

The reviewers and AC note that the strength of the paper includes a) an interesting compression algorithm of neural networks with provable guarantees (under some assumptions), b) solid experimental comparison with the existing *matrix sparsification* algorithms. The AC's main concern of the experimental part of the paper is that it doesn't outperform or match the performance of the "vanilla" neural network compression algorithms such as Han et al'15. The AC decided to suggest acceptance for the paper but also strongly encourage the paper to clarify the algorithms in comparison don't include state-of-the-art compression algorithms.